# Peroxisomal very long-chain fatty acid transport is targeted by herpesviruses and the antiviral host response

Isabelle Weinhofer[1✉], Agnieszka Buda[1], Markus Kunze[1], Zsofia Palfi[1], Matthäus Traunfellner[1], Sarah Hesse [1,2], Andrea Villoria-Gonzalez[1], Jörg Hofmann[3], Simon Hametner[4], Günther Regelsberger[4], Ann B. Moser[5], Florian Eichler[6], Stephan Kemp [7], Jan Bauer [8], Jörn-Sven Kühl[9], Sonja Forss-Petter[1] & Johannes Berger [1✉]

Very long-chain fatty acids (VLCFA) are critical for human cytomegalovirus replication and accumulate upon infection. Here, we used Epstein-Barr virus (EBV) infection of human B cells to elucidate how herpesviruses target VLCFA metabolism. Gene expression profiling revealed that, despite a general induction of peroxisome-related genes, EBV early infection decreased expression of the peroxisomal VLCFA transporters ABCD1 and ABCD2, thus impairing VLCFA degradation. The mechanism underlying ABCD1 and ABCD2 repression involved RNA interference by the EBV-induced microRNAs miR-9-5p and miR-155, respectively, causing significantly increased VLCFA levels. Treatment with 25-hydroxycholesterol, an antiviral innate immune modulator produced by macrophages, restored ABCD1 expression and reduced VLCFA accumulation in EBV-infected B-lymphocytes, and, upon lytic reactivation, reduced virus production in control but not ABCD1-deficient cells. Finally, also other herpesviruses and coronaviruses target ABCD1 expression. Because viral infection might trigger neuroinflammation in X-linked adrenoleukodystrophy (X-ALD, inherited ABCD1 deficiency), we explored a possible link between EBV infection and cerebral X-ALD. However, neither immunohistochemistry of post-mortem brains nor analysis of EBV seropositivity in 35 X-ALD children supported involvement of EBV in the onset of neuroinflammation. Collectively, our findings indicate a previously unrecognized, pivotal role of ABCD1 in viral infection and host defence, prompting consideration of other viral triggers in cerebral X-ALD.

[1] Department of Pathobiology of the Nervous System, Center for Brain Research, Medical University of Vienna, Vienna, Austria. [2] Institute of Molecular, Cell and Systems Biology, University of Glasgow, Glasgow, UK. [3] Institute of Virology, Charité - Universitätsmedizin Berlin, Berlin, Germany. [4] Division of Neuropathology and Neurochemistry, Department of Neurology, Medical University of Vienna, Vienna, Austria. [5] Department of Neurogenetics, Hugo W. Moser Research Institute at Kennedy Krieger, Kennedy Krieger Institute, Baltimore, MD, USA. [6] Department of Neurology, Harvard Medical School, Massachusetts General Hospital, Boston, MA, USA. [7] Genetic Metabolic Diseases, Department of Clinical Chemistry, Amsterdam University Medical Center, Amsterdam Neuroscience, Amsterdam Gastroenterology Endocrinology Metabolism, University of Amsterdam, Amsterdam, The Netherlands. [8] Department of Neuroimmunology, Center for Brain Research, Medical University of Vienna, Vienna, Austria. [9] Department of Pediatric Oncology, Hematology, and Hemostaseology, University Hospital Leipzig, Leipzig, Germany. ✉email: isabelle.weinhofer@meduniwien.ac.at; johannes.berger@meduniwien.ac.at

Human herpesviruses can directly cause viral encephalitis and have been associated with inflammatory demyelinating central nervous system (CNS) disorders like multiple sclerosis. A just released study by Bjornevik and colleagues revealed that infection with the herpesvirus Epstein-Barr virus (EBV) increases the risk for development of multiple sclerosis by 32-fold, thus providing evidence for EBV infection as the leading cause for this disorder[1]. However, it remains unclear whether herpesviruses represent an epiphenomenon that exacerbate the disease course or themselves act as a trigger for the onset of cerebral involvement[2]. Strengthening the link between viral infection and CNS disease is complicated by the ability of herpesviruses like EBV or human cytomegalovirus (HCMV) to enter latency following acute infection, making it difficult to trace the onset of neuropathology following viral infection. Treatment options are limited and vaccination is currently not available against EBV or HCMV. The development of alternative interventions requires further knowledge on the herpesvirus-mediated alterations of host cell metabolism and potential links to CNS involvement in genetically predisposed people.

Beyond immune dysregulation, herpesviruses hijack host lipid metabolism for the production of viral envelope components[3,4]. Infection with HCMV results in markedly elevated cellular levels of saturated very long-chain fatty acids (VLCFAs, ≥ C22), with C26:0 as the most abundant VLCFA of the viral envelope[5,6]. Importantly, interference with VLCFA synthesis led to viral particles with reduced infectivity[5,6]. Endogenous synthesis of long- and very long-chain fatty acids is accomplished by ER-embedded fatty acyl chain-selective enzymes (elongation of very long chain fatty acids, ELOVL). The ELOVL family comprises of seven members (ELOVL1-7) which catalyze the first, rate-limiting condensation step of the elongation cycle. In mammals, the concept of ELOVL1 being the major fatty acid elongase for production of C26:0 is based on expression studies in yeast, which are supported by almost undetectable amounts of C26:0 and C26:1 in both human and murine ELOVL1 deficiency[7–10]. In contrast to the synthesis, the degradation of VLCFAs by β-oxidation occurs in peroxisomes. Here, the import of saturated VLCFAs into the organelle by the transmembrane ATP-binding cassette (ABC) transporters ABCD1 and ABCD2 is rate-limiting and cell-type specific[11,12].

Accumulation of VLCFAs, in particular C26:0, is the biochemical characteristic of the inherited neurometabolic disorder X-linked adrenoleukodystrophy (X-ALD, OMIM #300100)[13,14]. Due to ABCD1 deficiency, the peroxisomal degradation of saturated VLCFAs is selectively impaired in X-ALD patients. The resulting build-up of these fatty acids in body fluids and tissues leads to pro-inflammatory skewing of innate immune cells[15]. Of note, X-ALD shows a striking phenotypic heterogeneity: while most patients surviving into adulthood develop a slowly progressing myeloneuropathy, involving spinal cord and peripheral nerves, more than half of the male patients, most often in childhood, experience severe, rapidly progressive inflammatory demyelination and axonal degeneration of the brain, termed cerebral ALD (CALD)[14,16,17]. Because no genotype–phenotype correlation exists in X-ALD, physical injury, epigenetic factors but also viral infections have been suggested to trigger the onset of CALD[18]. Historically, X-ALD was diagnosed by measurement of VLCFAs in patients' fibroblasts, and also B lymphocytes immortalized with EBV were attempted. Therefore, it was surprising that, in contrast to transformed B lymphocytes, primary B cells from X-ALD patients do not accumulate VLCFAs[12], which is probably due to relatively high expression of the related ABCD2 gene encoding an ABCD1 homologue with overlapping functions[19,20]. Apparently, EBV infection per se triggers VLCFA accumulation in B cells, possibly reflecting the importance of saturated VLCFAs for the EBV life cycle.

Here, we hypothesized that modulation of cellular VLCFA homeostasis could be a common strategy adopted by various (herpes)viruses to support their life cycle. Thus, using EBV as a paradigm for viral infections, the main objective of this study was to determine whether the peroxisomal import and degradation of VLCFAs is critical for viral infection and/or host defence. Our findings reveal an unexpected central role of ABCD1, a peroxisomal VLCFA transporter linked to the neuroinflammatory disorder X-ALD, in interfering with virus-mediated induction of VLCFA metabolism.

## Results

**EBV infection increases the cellular levels of saturated VLCFAs.** First, we determined to what extent EBV infection affects VLCFA metabolism in human B cells from healthy control donors by comparing the levels of the saturated VLCFAs of primary B cells with those of in vitro EBV-infected and transformed B lymphocytes. The levels of both C24:0 and C26:0 were significantly elevated upon EBV transformation of B cells (Fig. 1a, b). To elucidate whether loss of VLCFA degradation due to ABCD1 deficiency additionally impacts VLCFA levels, we next compared primary B cells and EBV-immortalized B lymphocytes from X-ALD patients and controls. Whereas the extent of induction of C24:0 levels was similar in control and X-ALD cells, the relative amount of C26:0 was further increased upon EBV transformation in X-ALD B lymphocytes, either as ratio to C22:0, as used in X-ALD diagnostics, or to C16:0 as standard reference fatty acid (Fig. 1a, b). To establish if VLCFA metabolism is affected early during EBV infection, we also assessed fatty acid levels before infected B cells are transformed by the virus into proliferating B lymphocytes. At 5 days post infection (dpi), we found markedly increased VLCFA species, including C26:0 (Fig. 1c). In contrast, no substantial change was observed for saturated long-chain fatty acids (C12–C20) such as C16:0, C18:0 or C20:0. Together, these results show increased VLCFA levels in response to EBV infection with more pronounced accumulation in ABCD1 deficiency, thus pointing to a role for ABCD1 in modulating VLCFA levels upon EBV infection.

**Despite a general induction of genes encoding peroxisomal proteins, the VLCFA transporters ABCD1 and ABCD2 are downregulated by EBV infection.** ABCD1 deficiency results in reduced peroxisomal degradation and, thus, accumulation of VLCFA in X-ALD cells. Therefore, we next assessed whether EBV infection and transformation of B cells affect expression of ABCD1 and its closest homologue, ABCD2, which has overlapping substrate specificity[19,20]. We found markedly reduced mRNA levels for both ABCD1 and ABCD2 in EBV-immortalized B lymphocytes when compared to non-infected primary B cells (Fig. 2a). These findings were confirmed for ABCD1 by reanalysis of data retrieved from Caliskan et al. (Supplementary Fig. 1)[21]. In contrast, we observed no significant alterations in mRNA levels of elongases catalyzing saturated LCFA and VLCFA synthesis such as ELOVL1 and ELOVL7 and even downregulated expression of ELOVL3 (Fig. 2b). In order to validate that the decreased ABCD1 and ABCD2 expression in B lymphocytes is accompanied by reduced rates of peroxisomal VLCFA degradation, we performed β-oxidation assays using [C14]-labelled C26:0 and, as a control, C16:0, a substrate for mitochondrial LCFA oxidation. Compared to the activity in non-infected primary B cells, the rate of C26:0 degradation was significantly reduced upon EBV infection and transformation, whereas no significant difference was found for C16:0 degradation (Fig. 2c). Next, we sought to determine whether targeting of the peroxisomal import of VLCFAs by downregulating ABCD1 and ABCD2 occurs early upon EBV infection before cells are transformed by the virus into B lymphocytes.

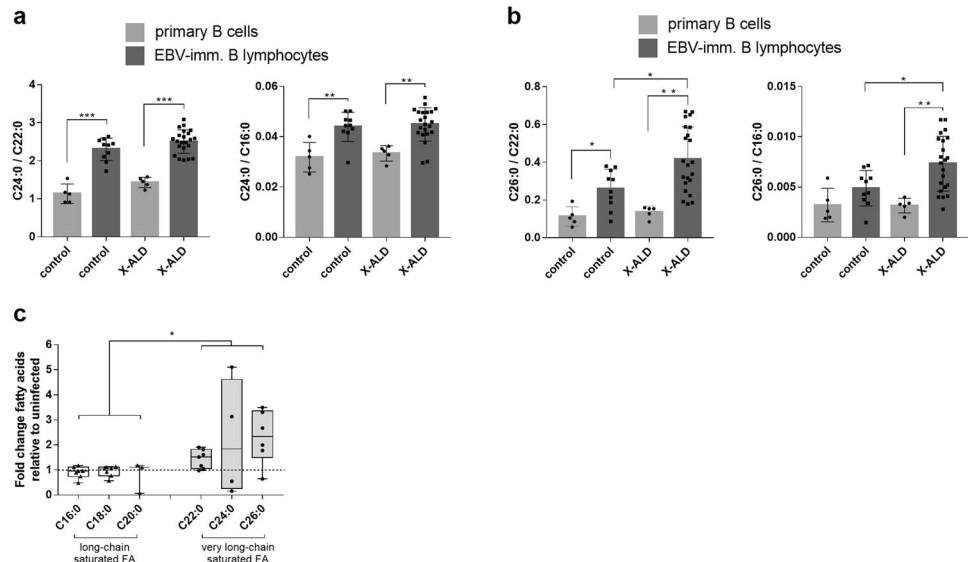

**Fig. 1 EBV infection increases VLCFA levels in B cells. a, b** The fatty acid concentrations of C26:0, C24:0, C22:0, and C16:0 were determined by GC-MS in primary B cells isolated from the blood of healthy controls ($n = 5$) and X-ALD patients ($n = 5$) as well as in in vitro EBV infected and immortalized B lymphocytes from controls ($n = 10$) or X-ALD patients ($n = 22$). The relative amounts of C24:0 (**a**) and C26:0 (**b**) displayed either as ratio to C22:0 or C16:0 are shown. The bar graphs show mean ± S.D. of the indicated values; $*P \leq 0.05$, $**P \leq 0.01$, $***P \leq 0.001$ (two-tailed unpaired Student's $t$-test, correction for multiple testing by Bonferroni adjustment). **c** The levels of saturated long-chain and very long-chain ($\geq$C22) fatty acids (FA) were determined by ESI-MS 5 days post infection of primary B cells from 7 healthy donors with EBV in vitro (MOI = 10). For C20:0, C24:0, and C26:0, the quantification was limited to $n = 3$, $n = 4$, and $n = 6$, respectively, due to low abundance and associated detection limits of these fatty acids. The data are depicted as boxplots (median ± interquartile range). $*P \leq 0.05$, two-tailed nested Student's $t$-test.

Therefore, we infected human primary B cells with EBV and analyzed *ABCD1* and *ABCD2* expression by RT-qPCR before and early after infection. Already at both 1 and 2 dpi, we observed reductions of *ABCD1* and *ABCD2* mRNA levels by about 80 and 50%, respectively (Fig. 2d), thus lending further support to the concept that EBV prevents peroxisomal VLCFA import.

To address how EBV infection generally impacts peroxisomes, we retrieved data from a time-resolved transcriptomic study by Mrozek-Gorska and collaborators, who investigated the early interactions between EBV and primary B cells[22]. By reanalyzing this data set with focus on peroxisomes and VLCFA metabolism, we found that EBV infection profoundly changed the expression of most genes related to peroxisomes starting early, at 1 dpi with a peak around 3 dpi (Fig. 2e). Opposing this overall upregulation of peroxisomal genes, the mRNA levels of the peroxisomal VLCFA transporters *ABCD1* and *ABCD2* were downregulated early upon infection, thus confirming our own results (Fig. 2e). Of note, however, the data from Mrozek-Gorska et al. revealed only a transient decrease in *ABCD1* and *ABCD2* expression with normalized levels by 4 dpi. This contrasts with our results and for *ABCD1* with the data retrieved from Caliskan et al.[21] (Supplementary Fig. 1) showing sustained repression in EBV-infected and immortalized B lymphocytes. Despite the accumulation of VLCFAs upon EBV infection and the suppression of both *ABCD1* and *ABCD2*, the expression of genes encoding enzymes participating in the peroxisomal β-oxidation of VLCFAs (*acyl-CoA oxidase 1, ACOX1*; *hydroxysteroid 17-beta dehydrogenase 4, HSD17B4* and *acetyl-CoA acyltransferase 1, ACAA1/thiolase*) were upregulated (Fig. 2e). In accordance with the general induction of peroxisomal genes, also mRNAs encoding enzymes involved in the peroxisomal steps of plasmalogen synthesis (*fatty acyl-CoA reductase 1, FAR1*; *glyceronephosphate O-acetyltransferase, GNPAT* and *alkylglycerone phosphate synthase, AGPS*) were transiently upregulated with a peak at 3 dpi (Fig. 2e). To reveal whether the transient induction of genes involved in ether lipid synthesis is still reflected by increased levels of plasmalogens at 5

dpi (Supplementary Fig. 2), we measured the total amounts of C16:0 and C18:0-containing plasmalogen subspecies as dimethyl-lacetal (DMA) derivatives. In contrast to the increased VLCFA levels at this time point (Fig. 1c), we did not observe significant alterations in total plasmalogen levels 5 dpi. Concomitant to the interference with VLCFA degradation, the RNA-Seq data[22] also pointed to a time-dependent early increase in VLCFA synthesis around 2–3 dpi, as indicated by a transient but robust induction of *ELOVL1* (Fig. 2f). In good agreement with the transcriptomic results, data retrieved and reanalyzed from a temporal proteomic data set, obtained during EBV infection of primary human B cells by Wang and colleagues[23], confirmed the general upregulation of proteins involved in β-oxidation, plasmalogen and VLCFA synthesis (ELOVL1) with concurrent downregulation of ABCD1 (Supplementary Fig. 3a). In our own sample set, we validated the markedly decreased ABCD1 protein levels at 2 and 5 dpi by Western blot analysis (Supplementary Fig. 3b).

**_ABCD1_ and _ABCD2_ are targeted by EBV-induced miRNAs.** Next, we set out to identify the molecular mechanism by which EBV infection could lead to reduced *ABCD1* and *ABCD2* expression. Small non-coding RNAs are prominent tools for viruses to manipulate both cellular and viral gene expression. Such microRNAs (miRNAs, miRs) can induce degradation of targeted mRNAs and translational silencing to adapt the host's gene expression profile for viral growth[24,25]. To elucidate if *ABCD1* and *ABCD2* mRNAs are targets of EBV-induced miRNAs, we applied several public miRNA target prediction algorithms to localize potential target sites within the 3′-untranslated regions (UTRs) of *ABCD1* and *ABCD2*. We identified two putative miR-9-5p binding sites in the 3′-UTR of *ABCD1* and three putative miR-155 binding sites in the 3′-UTR of *ABCD2* (Fig. 3a, e and Supplementary Tables 1, 2). We observed both miR-9-5p and miR-155 (cellular miRNAs) being strongly upregulated upon EBV infection and immortalization of primary human B cells (Fig. 3b, f), thus confirming previous reports[26,27]. To demonstrate an effect of miR-9-5p and miR-155 on the predicted

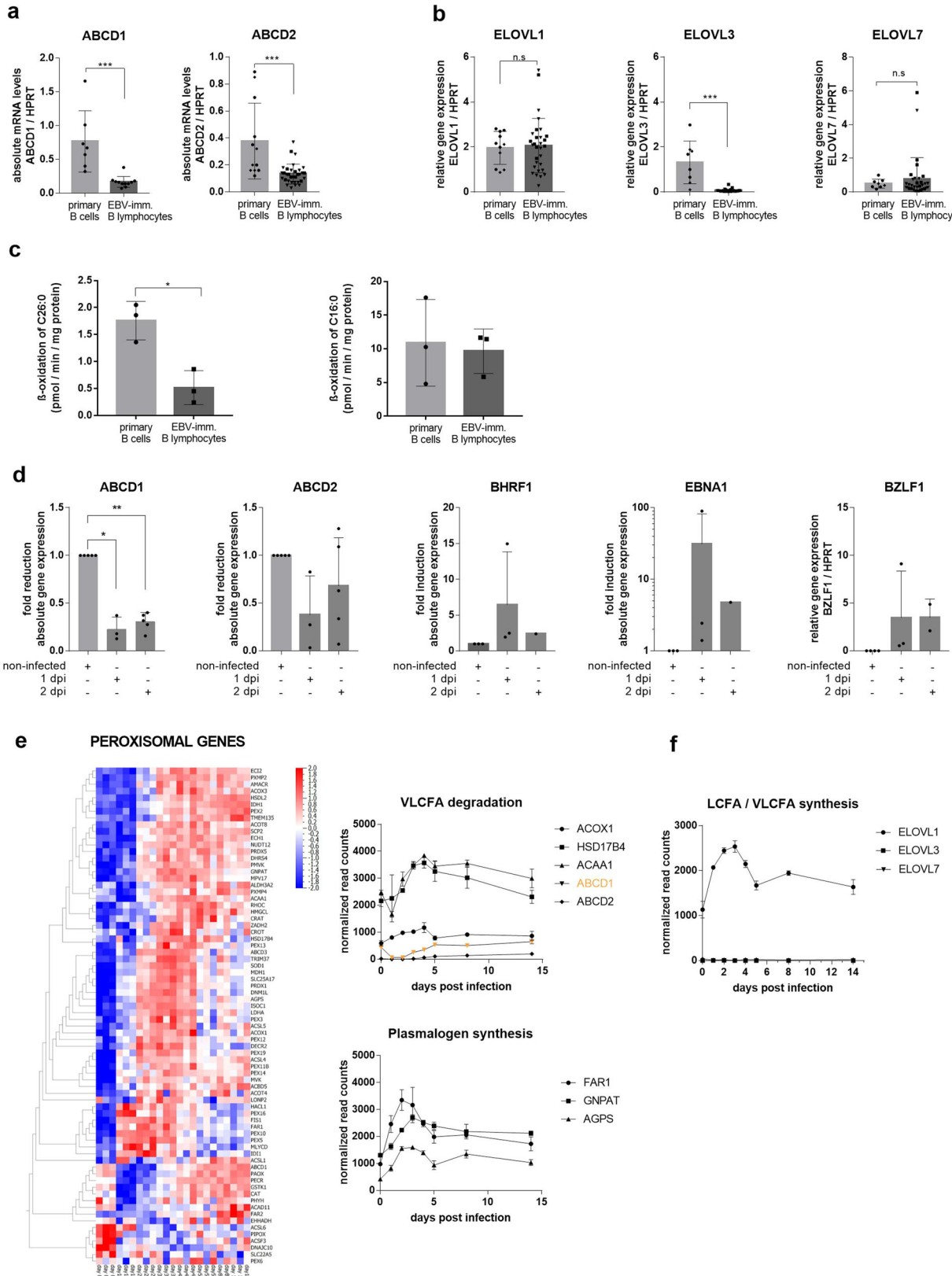

target sites of the human *ABCD1* and *ABCD2* mRNA, respectively, we employed dual-luciferase reporter assays using HEK-293 cells, a human cell line known to respond to miRNA mimics. Co-transfection of the *ABCD1-* or *ABCD2-*3′-UTR reporter constructs with either miR-9-5p or miR-155 mimics, respectively, resulted in significantly decreased luciferase activity when compared to non-

targeting miRNA negative controls. Moreover, a non-related 3′-UTR sequence of the β-actin gene was not affected by these mimics. To verify a direct interaction of these miRNAs and the predicted target sequences, we performed in vitro mutagenesis of the binding sites and observed that mutations in the second target site (M-TS2) of the *ABCD1-*3′-UTR were sufficient to ablate the effect of the miR-9-5p

**Fig. 2 Expression dynamics of genes related to peroxisomes and VLCFA synthesis during EBV infection. a**, **b** Comparison of primary B cells and EBV-immortalized B lymphocytes. RNA was isolated from primary B cells (healthy controls, $n = 4$–7, black circles; X-ALD patients, $n = 4$–5, black rhombs) and EBV-infected and immortalized B lymphocytes (healthy controls, $n = 10$–11, black squares; X-ALD patients, $n = 20$–22, black inverted triangles) and RT-qPCR was carried out for *ABCD1*, *ABCD2*, *ELOVL1*, *ELOVL3*, and *ELOVL7*. Data were normalized to *HPRT*. The bar graphs show mean ± S.D. of the indicated values. For statistical analysis, two-tailed unpaired Student´s *t*-test was used (***$P \leq 0.001$; n.s. = non significant). **c** Degradation rates of C26:0 (left) and C16:0 (right) by peroxisomal and mitochondrial β-oxidation, respectively, were determined in primary B cells ($n = 3$) and EBV-infected and immortalized B lymphocytes ($n = 3$) from healthy controls. The bar graphs show mean ± S.D. of the indicated values. For statistical analysis, two-tailed unpaired Student´s *t*-test was used (*$P \leq 0.05$). **d** Expression profile of primary B cells before and after EBV infection. RNA was isolated from primary B cells of healthy donors ($n = 3$–5) before and at one or two days post infection (dpi) with EBV in vitro. RT-qPCR was carried out for the *ABCD1* and *ABCD2* genes with viral *EBNA1*, *BHRF1*, and *BZLF1* serving as controls for successful infection. *HPRT* was used for normalization purposes. The bar graphs show mean ± S.D. of the indicated values. For statistical analysis, two-tailed ratio paired Student´s *t*-test was used (*$P \leq 0.05$, **$P \leq 0.01$). **e**, **f** Time-resolved RNA-Seq data from B cells, isolated from three healthy donors and infected in vitro with EBV, was retrieved from Mrozek-Gorska et al.[22]. Data normalized by Mrozek-Gorska and colleagues for sequencing depth using size factors were imported into the Qlucore Omics Explorer, where log-transformation and scaling with Z-score was carried out. Heatmap and graphs of peroxisome-related genes (**e**) and elongase encoding genes involved in saturated LCFA and VLCFA synthesis (**f**) at different time points following infection. Depicted data points are means of three donors ± S.D.

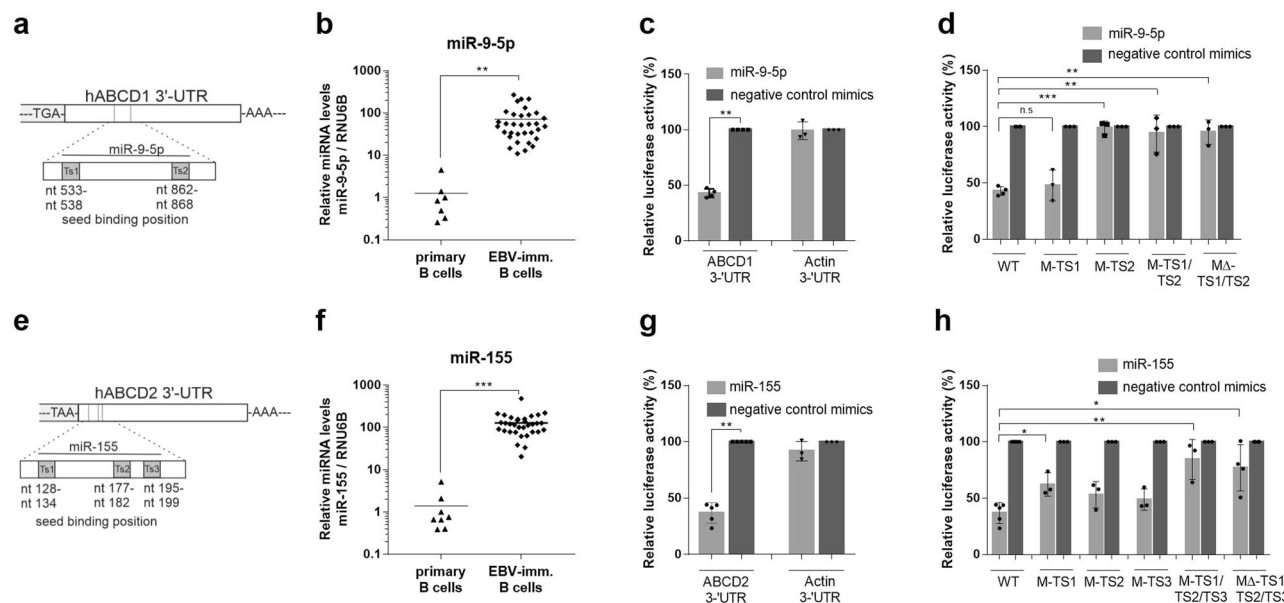

**Fig. 3 The peroxisomal VLCFA transporters ABCD1 and ABCD2 are targets of EBV-induced miRNAs. a**, **e** Schematic diagrams of the 3′-UTR of the human ABCD1 (1044 bp) and ABCD2 (1496 bp) cDNAs. The relative locations of the putative miR-9-5p (TS1: nt 533–538, TS2: nt 862–868) and miR-155 (TS1: nt 128–134, TS2: nt 177–182, and TS3:195–199) target sites are shown as grey boxes. **b**, **f** RT-qPCR analysis of miR-9-5p and miR-155 expression in primary B cells ($n = 7$–8) and EBV-immortalized B lymphocytes ($n = 33$). The miRNA levels were normalized to the small non-coding RNA, RNU6B, as unregulated reference. The mean is indicated by a horizontal line. **c**, **d**, **g**, **h** Luciferase reporter assays in HEK-293 cells harbouring ABCD1- (**c**) or ABCD2- (**g**) 3′-UTR constructs and a β-actin-3′-UTR (negative control) construct co-transfected with either miR-9-5p, miR-155 or non-targeting negative control mimics as indicated. The data is displayed as percentage of Renilla/Firefly luciferase activity ratio (mean ± S.D.) relative to that of cells transfected with negative control mimics. Each data point represents an independent transfection experiment. In the mutated ABCD1-3′-UTR constructs M-TS1, MT-S2, M-TS1/TS2, and MΔ-TS1/TS2 (**d**) or ABCD2-3′-UTR constructs M-TS1, M-TS2, M-TS3, M-TS1/TS2/TS3 and MΔ-TS1/TS2/TS3 (**h**), the putative miR-9-5p or miR-155 binding sites were either in vitro mutagenized (M) or deleted (MΔ). *$p \leq 0.05$, **$p \leq 0.01$, ***$p \leq 0.001$ (two-tailed unpaired Student's *t*-test, correction for multiple testing by Bonferroni adjustment).

mimic on the reporter construct (Fig. 3c, d) whereas the concomitant introduction of mutations into all three miR-155 binding sites in the *ABCD2*-3′-UTR (M-TS1/TS2/TS3) or their deletion reduced the effect size of the miR-155 mimics to about 10% (Fig. 3g, h). Collectively, these results further validate ABCD1 and ABCD2 as direct functional targets of EBV-induced miR-9-5p and miR-155 and indicate that upon EBV infection, the peroxisomal degradation of VLCFAs may be evaded through miRNA-mediated repression of *ABCD1* and *ABCD2*.

**EBV serum positivity is not elevated in children with inflammatory cerebral X-ALD.** Environmental factors are thought to play a role in the clinical manifestations of X-ALD. In these

patients, the inherited loss of functional ABCD1 leads to significant accumulation of VLCFAs in various tissues and body fluids. To test the hypothesis that EBV constitutes a trigger for the onset of inflammatory cerebral ALD, we investigated the presence of EBV seropositivity in a group of 35 children with MRI-confirmed CALD lesions (age range 4–15 years, median age = 8 years). Our analysis revealed similar incidences in the X-ALD (63%) and the age-matched control groups (70%) consisting of 285 children (age range 4–15 years, median age = 8 years) with conditions unrelated to X-ALD (Supplementary Fig. 4a–c). Parallel determination of the EBV viral-capsid antigen (VCA) IgG, Epstein-Barr nuclear antigen (EBNA) IgG and EBV IgM, enabling the discrimination of different phases of the EBV

infection, indicated no association of a recent EBV infection with onset of neuroinflammation in EBV-positive CALD patients (Supplementary Fig. 4d). Finally, by using immunohistochemistry for detection of EBV latent membrane protein 1 (LMP1) and in situ hybridization for the EBV-associated small non-coding RNA EBER, we were unable to identify EBV positivity in post-mortem brain tissue of four CALD cases (Supplementary Fig. 4e, f). Based on these observations, we conclude that in the investigated cohort of CALD patients, the onset of neuroinflammation could not be correlated to a previous or acute EBV infection.

**ABCD1 repression is shared by different herpes- and coronaviruses.** To elucidate whether additional herpesviruses or even other virus families are associated with repression of *ABCD1*, we queried publicly available transcriptomic data sets of host cell expression changes upon infection with herpesviruses including herpes simplex virus HSV-1 (HHV-1)[28,29], varicella-zoster virus VZV (HHV-3)[30], EBV (HHV-4)[31], CMV (HHV-5)[32], roseolovirus (HHV-6,7)[33], Kaposi´s sarcoma-associated herpesvirus (KSHV/HHV-8) and coronaviruses including the Middle East respiratory syndrome coronavirus (MERS-CoV)[34,35], the severe acute respiratory syndrome coronavirus 1 (SARS-CoV-1)[36] and the novel SARS-CoV-2[37]. By mining these data sets, we found that downregulated expression of *ABCD1* was a feature shared by different members of the herpes- and coronavirus families (Fig. 4a–i and Supplementary Fig. 5). Of note, by analysis of data characterizing the transcriptional signature of the host inflammatory response to SARS-CoV-2[38], a significant upregulation of *ABCD1* expression in peripheral blood mononuclear cells (PBMCs) of COVID-19 patients was observed (Fig. 4j), possibly hinting at a role of ABCD1 also in the antiviral host response ensuing from infection.

**The antiviral metabolite 25-hydroxycholesterol reverts EBV-induced VLCFA accumulation and blocks EBV replication in vitro.** Accordingly, we next asked whether the antiviral host response had evolved a mechanism to counteract the down-regulation of *ABCD1* expression upon virus infection. Previous reports have demonstrated that the natural cholesterol metabolite 25-hydroxycholesterol (25-HC), an immune modulator secreted by macrophages during the host response, confers broad antiviral activity by blocking entry, replication and production of infectious viruses of diverse enveloped viruses[39–42] and, as recently described, including also *de novo* EBV infection of B cells[43]. Here, we hypothesized that 25-HC might also interfere with EBV lytic replication, possibly by restoring *ABCD1* expression and VLCFA homeostasis in EBV-infected cells. To test this idea, we first supplemented EBV-infected, immortalized healthy control B lymphocytes with 25-HC and analyzed the impact on VLCFA metabolism. After 48 h we found significantly elevated peroxisomal β-oxidation of C26:0 (Fig. 5a), with subsequent reduction of both C26:0 and C24:0 VLCFA levels upon 3 weeks of treatment (Fig. 5b). To investigate whether 25-HC opposes the virus-induced downregulation of *ABCD1*, thus preventing VLCFA accumulation, we analyzed ABCD1 mRNA and protein levels after 24 and 48 h of treatment, respectively, of the EBV-immortalized B lymphocytes and observed induced ABCD1 expression in all tested cell lines (Fig. 5c, d). Conversely, 25-HC repressed *ELOVL1* mRNA levels (Fig. 5c), confirming a previous observation in human fibroblasts[44]. In contrast, expression of *ABCD2* and genes involved in plasmalogen synthesis were unaffected by 25-HC (Supplementary Fig. 6). We next tested whether a 1 or 3 week 25-HC treatment interferes with EBV reproduction upon lytic reactivation of EBV-immortalized B lymphocytes and if such an impedance would involve ABCD1 function. In healthy

control B lymphocytes, the decreased VLCFA levels that we observed after 3 weeks of 25-HC treatment correlated with a reduced capacity of EBV for lytic reactivation, as indicated by significantly lower virus release upon pharmacological induction (Fig. 5e). In contrast, 25-HC treatment did not significantly interfere with EBV reproduction in the X-ALD-derived B lymphocytes lacking ABCD1 (Fig. 5e). Collectively, these data show that the antiviral mediator 25-HC restores ABCD1 expression in B lymphocytes, resulting in reduced VLCFA levels and an ABCD1-dependent decrease in EBV production (Fig. 5f). Finally, we tested whether the impact on *ABCD1* and *ELOVL1* expression is shared by other compounds known to interfere with EBV replication. Accordingly, we analyzed the expression of *ABCD1*, *ELOVL1* and also *ABCD2* in EBV-immortalized healthy control B lymphocytes treated with the EBV lytic cycle inhibitor valpromide for 24 h. Our data revealed that similar to 25-HC, valpromide significantly upregulated *ABCD1* mRNA levels and additionally stimulated *ABCD2* expression in EBV-immortalized B lymphocytes. However, in contrast to 25-HC, valpromide had no significant effect on *ELOVL1* transcription (Supplementary Fig. 7).

## Discussion

Viruses have developed highly sophisticated strategies to exploit the host lipid metabolism to promote their replication. Our present study revealed that instructing host cells to accumulate VLCFAs apparently is a common strategy, adopted not only by HCMV but also by EBV and possibly other herpesviruses to support their propagation. Whereas previous reports demonstrated that VLCFAs are enriched upon HCMV infection by targeting the fatty acid elongation process[5,6], we here complement this knowledge by showing that several herpesviruses, including EBV and HCMV, and coronaviruses target *ABCD1*, the main transporter mediating peroxisomal degradation of saturated VLCFAs. Using EBV infection of B cells as a paradigm for herpesvirus infection, our observations show that despite a general induction of peroxisome-related genes, viral infection simultaneously interferes with the β-oxidation of VLCFAs by preventing their peroxisomal import (Fig. 5f). Interestingly, we discovered that EBV induces host-derived miRNAs including miR-9-5p and miR-155 that repress *ABCD1* and *ABCD2*, two related peroxisomal fatty acid transporters with overlapping substrate specificity but distinct expression profiles. With miR-9-5p also being part of the HCMV host-virus crosstalk and *ABCD1* also being targeted by the HCMV miRNA hcmv-miR-US29-3p[45], both the targeting of as well as the molecular mechanism of *ABCD1* repression involving miRNAs might be conserved between different viruses. Lending support to this concept, a screen for host-virus interactions of the mirTar database (https://mcube.nju.edu.cn/jwang/mirTar/docs/mirTar/) revealed 61 putative interactions of virus miRNAs with *ABCD1* and 12 with *ABCD2*, including predicted binding sites for the EBV miRNAs ebv-miR-BART10-5p and ebv-miR-BART1-5p in the *ABCD1* and *ABCD2* mRNAs, respectively. Further studies are needed to confirm the functionality of these predicted host-virus interactions.

The importance of VLCFAs for a variety of enveloped viruses is further corroborated by our discovery of *ABCD1* as a common target. VLCFAs are thought to influence membrane properties such as curvature and permeability[46,47]. This feature of VLCFA-containing membrane lipids for cell-shape regulation could be especially important for budding of virions[46] but might also be relevant for inducing latency and/or modulating the host immune response. Furthermore, not only viruses but also the rice blast fungus *Magnaporthe oryzae* seem to depend on VLCFAs for pathogenicity[48]. In this report, He and co-workers show that the

## a - i *in vitro* virus infection

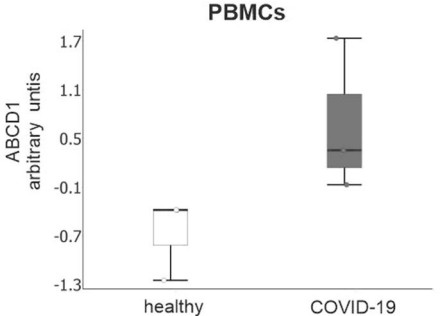

## j   host inflammatory response to SARS-CoV-2

fungus uses VLCFAs to recruit filament-forming septins to curved plasma membranes, thereby promoting the formation of a penetration peg required for host infection. The link between VLCFAs and host-pathogen interaction and virulence raises the question whether host cells have co-evolved mechanisms that interfere with the requirement of pathogens for VLCFAs. Here, the elongation of LCFAs to VLCFAs by the enzyme ELOVL1 and the degradation of VLCFAs by peroxisomal β-oxidation may represent critical steps for VLCFA homeostasis and availability during viral infection. Our finding that the antiviral cholesterol metabolite 25-HC promotes cellular VLCFA degradation by positive modulation of *ABCD1* expression with simultaneous downregulation of *ELOVL1* to curb VLCFA synthesis lends support to this concept. Of note, the ability of 25-HC to stimulate

**Fig. 4 ABCD1 expression is targeted by herpes- and coronaviruses.** Transcriptomics datasets from herpes- or coronavirus in vitro-infected human cells (−, mock-infected; +, virus infected) or from peripheral blood mononuclear cells (PBMCs) derived from COVID-19 patients were retrieved from Gene Expression Omnibus database files. The data are depicted as boxplots (median ± interquartile range). **a** HSV-1 infected primary fibroblasts 9 h post infection (pi) (MOI = 10, two biological replicates, GSE129582); **b** VZV infected melanoma cells, 36 h pi (two biological replicates, GSE85493); **c** EBV infected gastric cancer cells, 48 h pi (5 replicates, GSE135644); **d** HCMV infected lung fibroblasts, 72 h pi (MOI = 10, two biological replicates, GSE99454); **e** roseolovirus infected T lymphoblastoid cells, 72 h pi (MOI = 20, one biological replicate, GSE149808); **f** KSHV infected Tert-immortalized microvascular endothelial cells, 48 h pi (three replicates, GSE27136); **g** MERS-CoV infected bronchial epithelial cells, 24 h pi (MOI = 5, three replicates, GSE45042); **h** SARS-CoV-1 infected bronchial epithelial cells, 3 h pi (MOI = 5, three replicates, GSE33267); **i** SARS-CoV-2 infected iPSC-cardiomyocytes, 72 h pi (MOI = 0.1, three replicates, GSE150392); **j** PBMCs derived from three COVID-19 patients and three controls (CRA002390, https://bigd.big.ac.cn/). Arbitrary units on the y-axis represent normalized variable values that were obtained using the default settings to Z-score normalization in the Qlucore Software 3.5 (mean zero and standard deviation 1). For data sets including longitudinal sampling, time courses are shown in Supplementary Fig. 5. Log2 fold changes and P-values are shown in Supplementary Table 3.

ABCD1 expression was shared by the EBV lytic cycle inhibitor valpromide, possibly indicating a more general mechanism in targeting ABCD1 to interfere with EBV replication.

In general, the mechanisms underlying the antiviral properties of 25-HC are still incompletely understood. One hypothesis is that upon stimulated secretion through cells of the innate immune response, 25-HC impairs the fusion of viral envelope and cell membrane by changing the properties of host membranes. Indeed, 25-HC lacks the membrane-ordering and packing capacities of cholesterol, thus membrane fluidity is altered upon its incorporation[49,50]. Because the VLCFA C26:0 has similar hydrophobicity and dissociation rate from membranes as cholesterol[51,52], it seems plausible that limiting also C26:0-containing lipids is necessary to make membranes less favourable for viral entry and/or release. Thus, one could hypothesize that the lack of efficacy of 25-HC on stimulating ABCD1 expression and thus, the inability to induce VLCFA transport and degradation in the context of ABCD1 deficiency, might interfere with the capacity of 25-HC to regulate the VLCFA-content in specific membrane domains required for the release of EBV virions from infected cells. Future research needs to address in detail the underlying mechanism for how the lack of ABCD1 interferes with the ability of 25-HC to prevent EBV replication. With our data linking ABCD1 function to antiviral properties of 25-HC in the context of EBV replication, the question raised whether ABCD1 expression might be associated with activation of the EBV lytic cycle in infected B cells. However, using an interactive worksheet previously published by Ersing and colleagues, who reported a temporal proteomic map of EBV B cell lytic replication[53], we found no support linking a change in ABCD1 protein levels to induction of EBV lytic replication.

The opposing effects on ABCD1 being induced for VLCFA degradation and ELOVL1 being inhibited for reduced synthesis of VLCFA, consequently decreasing the VLCFA levels, adds yet another modality to how 25-HC perturbs viral entry, replication, and release. Although 25-HC is effective in vitro against a wide range of viruses including SARS-CoV-2[42] and, just recently reported, also against de novo EBV infection by suppressing key viral genes and interfering with viral induced proliferation[43], the diversity of 25-HC functions in cells, ranging from membrane integrity to cell signalling and regulation of gene expression through the nuclear liver X receptor (LXR), hampers the direct use of this oxysterol as a specific therapeutic agent against viruses. Accordingly, the concept of limiting VLCFAs by compounds mediating increased peroxisomal transport and degradation of VLCFAs could open up a new alternative research path for development of strategies to suppress virus uptake and release.

It was tempting to speculate that the inherent ABCD1 deficiency in X-ALD patients would confer a higher risk for EBV infection and, in concert with a pro-inflammatory skewed innate immune response[15], triggers the onset of the severe CALD disease course.

However, we did not detect a higher prevalence of EBV seropositivity in childhood CALD patients than in controls of similar age with conditions unrelated to X-ALD. Of note, the determined prevalence of EBV infection in our cohort of control children aged 4–15 years was comparable to rates previously observed for children within this age group in Berlin/Germany[54], which was higher than the incidence reported in a U.S. study[55]. Further, we could not reveal an association between recent EBV infections and onset of neuroinflammation in the investigated cohort of CALD patients. Together, these findings do not support the hypothesis of EBV being a trigger of cerebral involvement in these children. However, our observation that ABCD1 is targeted upon infection by a variety of viruses, calls for consideration also of other herpesviruses, or other enveloped virus families like coronaviruses, as candidate environmental factors that may contribute to CALD onset.

All together our data revealing that the peroxisomal VLCFA transporter ABCD1, a protein associated with X-linked adrenoleukodystrophy, and VLCFAs are targeted by both viral and antiviral host-cell strategies may fuel further research aimed to clarify the complex coevolution of pathogens and their hosts and potential links to neuroinflammation.

## Methods

**X-ALD patients and healthy volunteers.** The study included peripheral blood samples from 5 adult X-ALD patients (median age = 38 years, SD = 4.12) and 20 healthy volunteers (median age = 37 years, SD = 10.64) for isolation of B cells. X-ALD patients displayed clinical symptoms of axonopathy in the spinal cord but no signs of cerebral involvement (CALD) at brain MRI. The accumulation of saturated VLCFAs in the blood of X-ALD patients was confirmed by measurement of the total amount of the fatty acids C26:0, C24:0, C22:0 and, for normalization, C16:0 by using gas chromatography–mass spectrometry (GC–MS). The study was approved by the Ethical Committee of the Medical University of Vienna (EK1462/2014) and informed consent was obtained from participating X-ALD patients and healthy volunteers.

**EBV-specific serology.** EBV-specific antibodies were determined in samples from children with CALD (n = 35, median age = 8) and age-matched controls (n = 285, median age = 8 years). The serologic testing was performed by detection of IgG and IgM directed to the viral capsid protein (VCA, antigen used: p18; REF 310510 and REF 310500, respectively, DiaSorin S.p.A., Saluggia, Italy) and IgG to the EB nuclear antigen-1, applying LIAISON® EBNA IgG (REF 310520, DiaSorin S.p.A., Saluggia, Italy). Chemiluminescent immunoassays were carried out using a LIAISON XL (DiaSorin S.p.A., Saluggia, Italy). Parallel determination of VCA IgG, EBNA IgG and EBV IgM levels enabled the discrimination among different phases of the EBV infection. EBV serology for CALD patients was done as part of their standard transplant routine, for controls within their clinical diagnostic work-up. Patient consent for this retrospective study was waived by the local institutional review board (Ethical Committee Charité), as no patient was contacted and only pseudonymized information from existing medical records was used. Retrospective analysis of the blood samples was approved by the Ethical Committee of the Medical University of Vienna (EK1613/2019).

**Immunohistochemistry.** Immunohistochemical analysis was performed on formalin-fixed, paraffin-embedded sections using primary antibodies for CD20 (Thermo scientific, MS-340, 1:100) and EBV-LMP (Dakocytomation, M0897, 1:60). Visualization of bound primary antibody was performed by incubation with

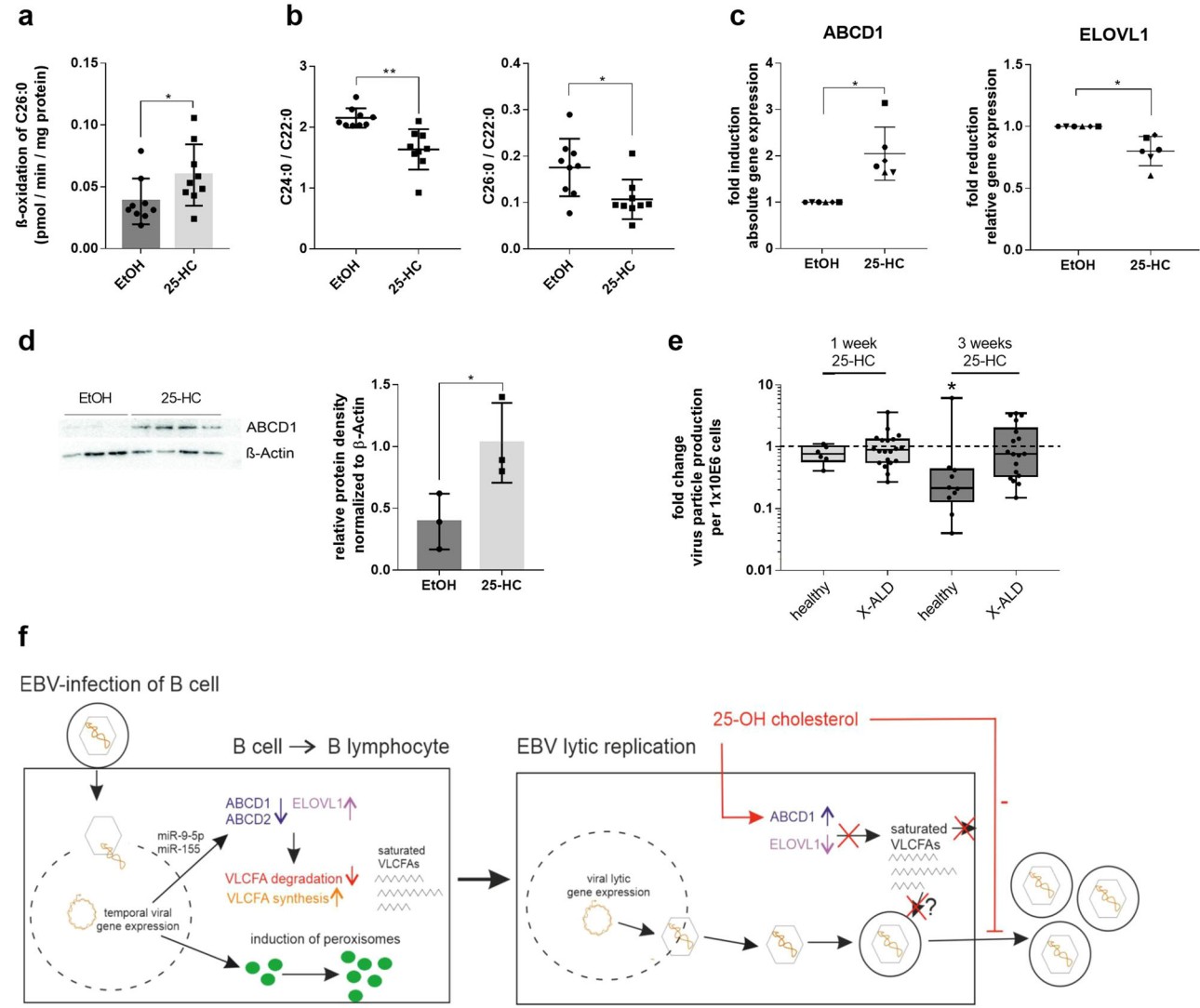

**Fig. 5 The antiviral mediator 25-hydroxycholesterol lowers cellular VLCFA levels and interferes with EBV reproduction.** In vitro EBV-infected and immortalized B lymphocytes from healthy controls were supplemented with either 25-HC (2 μM) or vehicle (ethanol, EtOH). **a** After 48 h, the degradation of C26:0 was determined and displayed as the C26:0 β-oxidation rate normalized to protein content ($n = 9$). The bar graphs show mean ± S.D. of the indicated values. **b** The cellular amounts of C22:0, C24:0 and C26:0 were determined by GC-MS after 3 weeks of treatment. The levels of C24:0 and C26:0 are displayed as ratio to C22:0 ($n = 9$). The mean is indicated by a horizontal line ± S.D. **c** RNA was isolated after 24 h and RT-qPCR was carried out for *ABCD1* and *ELOVL1* ($n = 6$). The mean is indicated by a horizontal line ± S.D. Data were normalized to *HPRT* mRNA levels and displayed as fold change compared to vehicle-treated cells. **d** Protein extracts for Western blot analysis were prepared after 48 h of supplementation with 25-HC or EtOH. The relative amount of ABCD1 protein was normalized to the β-actin level. One representative immunoblot from one B lymphocyte line with 3 (untreated) and 4 (25-HC treated) technical replicates is shown; quantification was conducted using three different B lymphocyte lines. The bar graphs show mean ± S.D. of the indicated values. Uncropped western blot images are shown in Supplementary Fig. 8. **e** EBV-immortalized B lymphocytes derived from X-ALD patients ($n = 18–20$) or healthy controls ($n = 6–10$) were treated with either 25-HC (2 μM) or EtOH for either 1 or 3 weeks before lytic virus replication was induced by PMA/sodium butyrate. EBV particle number was determined in the cell free supernatant by qPCR for *BALF5* in the EBV genome. The data are depicted as boxplots (median ± interquartile range). For statistical analysis, two-tailed paired Student's *t*-test was used (\**P* ≤ 0.05, \*\**P* ≤ 0.01). In (**c**) and (**e**), the raw values used to generate the fold-change display were used for statistical analysis. **f** Model of EBV targeting VLCFA metabolism and peroxisomes (left panel) and how 25-HC impedes viral reproduction (right panel). EBV infection of B cells results in rapid miRNA-mediated downregulation of the peroxisomal VLCFA importers ABCD1 and ABCD2. This occurs despite a general induction of peroxisome-related genes such as those encoding proteins involved in peroxisome proliferation and biogenesis (c.f. Fig. 2e), possibly indicating increased peroxisome numbers. The impaired peroxisomal degradation of VLCFAs but concurrent upregulation of VLCFA synthesis via the fatty acid elongase ELOVL1 leads to increased cellular VLCFA levels, possibly necessary to establish latency. Upon stimulation of EBV lytic replication, the antiviral metabolite 25-hydroxycholesterol interferes with EBV reproduction, among other, by opposite regulation of *ABCD1* (up) and *ELOVL1* (down), thus resulting in net VLCFA catabolism and preventing cellular accumulation of VLCFAs.

Biotin-labelled donkey anti-mouse secondary antibodies (#715-065-150, Jackson ImmunoResearch Laboratories, West Grove, PA) for 1 h at room temperature and detection with avidin-labelled peroxidase (Sigma, 1:150) and diaminobenzidine tetrachloride (DAB, Sigma). A hematoxylin counterstain was performed to visualize nuclei.

**EBER in situ hybridization**. EBER in situ hybridization was performed using a commercially available kit. Briefly, formalin-fixed, paraffin-embedded sections were hybridized with the Inform EBER probe (#05278660001, Roche Diagnostics) and the VENTANA ISH iView Blue Detection Kit was used for visualization (#05278511001, Roche Diagnostics). The staining procedures were performed in a

Ventana BenchMark ULTRA IHC/ISH System (Roche Diagnostics) according to guidelines from the supplier.

**Cells and cell culture**. Primary B cells were isolated from peripheral blood by Ficoll density-gradient centrifugation (PAN-Biotech) and positive selection for CD19+ cells using MACS microbeads and the LS column system (Miltenyi Biotec) according to the manufacturer's instructions. The purity of isolated primary B cells was determined using a mouse anti-human CD19 monoclonal antibody, PE conjugated (clone LT19, # 130-091-247, Miltenyi) and the PE-conjugated monoclonal antibody isotype control (#120-002-723, Miltenyi) by flow cytometry. Primary B cells isolated from the blood and EBV-immortalized B lymphocytes kindly provided by Dr. Ann Moser, Kennedy Krieger Institute, Baltimore, USA and Prof. Florian Eichler, Harvard Medical School, Boston, USA, were maintained in RPMI complete medium [RPMI 1640 supplemented with 2 mM L-glutamine, 100 μg/ml streptomycin, 100 U/ml penicillin (all Invitrogen) and 10% heat-inactivated fetal calf serum (Gibco Life Technologies)]. EBV-immortalized B lymphocytes were treated with 2 μM 25-HC or the solvent ethanol for the indicated time period. HEK-293 cells were grown in Dulbecco's modified Eagle's medium (DMEM; Sigma-Aldrich) supplemented as described above for RPMI. All cells were cultivated at 37 °C and 5% $CO_2$.

**Induction of EBV production and release in EBV-immortalized lymphocytes**. To induce EBV replication in immortalized B lymphocytes, $1.5 \times 10^6$ cells per ml were treated with 3 mM sodium butyrate and 20 ng/ml phorbol myristate acetate (PMA) for 4 days. The virus-containing supernatant was filtered through 0.45 μm membranes and the EBV particle number determined as the EBV genome copy number by qPCR of the EBV gene *BALF5* using the primers listed in Supplementary Table 4.

**EBV infection of primary B cells**. For EBV infection, primary B cells from healthy donors were incubated with EBV-containing supernatant derived from sodium butyrate/PMA stimulated B lymphocytes: approximately $1 \times 10^7$ EBV particles were added to $1 \times 10^6$ primary B cells at 37 °C for 3 h (multiplicity of infection, MOI = 10). After washing cells in new medium, the infected B cells were further incubated for up to 5 days before harvesting.

**Fatty acid and plasmalogen analysis**. The total amounts of saturated and mono-unsaturated fatty acids were determined using GC-MS (C16:0, C22:0, C24:0 and C26:0; Fig. 1a, b) and electrospray ionization mass spectrometry (ESI–MS; C16:0, C18:0, C20:0, C22:0, C22:1, C24:0, C24:1, C26:0 and C26:1; Fig. 1c) methods using deuterated internal standards[12,56]. For analysis of plasmalogen content, the total amounts of C16:0 and C18:0-containing plasmalogen subspecies were measured as dimethylacetal (DMA) derivatives[57].

**Peroxisomal β-oxidation of 1-$^{14}$C-labelled fatty acids**. [1-$^{14}$C]-palmitic acid (C16:0; ARC 0172A) and [1-$^{14}$C]-hexacosanoic acid (C26:0; ARC 1253) were obtained from American Radiolabeled Chemicals (St. Louis, MO, USA). Free fatty acids in ethanol were aliquoted into glass reaction tubes, dried under a stream of nitrogen and solubilized in 10 mg/ml α-cyclodextrin by ultrasonication. The reaction mix contained 4 μM of labelled fatty acids, 2 mg/ml α-cyclodextrin, 30 mM KCl, 8.5 mM ATP, 8.5 mM MgCl$_2$, 1 mM NAD$^+$, 0.17 mM FAD, 2.5 mM l-carnitine, 0.16 mM CoA, 0.5 mM malate, 0.2 mM EDTA, 1 mM DTT, 250 mM sucrose and 20 mM Tris-Cl, pH 8.0. Reactions were started by addition of $5 \times 10^6 – 2 \times 10^7$ cells, carried out for 1 h at 37 °C and stopped by addition of KOH and heating to 60 °C for 1 h. After protein precipitation by HClO$_4$, a Folch partition was carried out and $^{14}$C-acetate was determined in the aqueous phase by scintillation counting using the Perkin Elmer Tri-Carb 4910TR Scintillation Counter.

**RNA isolation and reverse transcription-coupled quantitative PCR (RT-qPCR)**. RNA was isolated from primary B cells and EBV-immortalized B lymphocytes using the RNeasy Mini Kit (Qiagen) or the mirVana™ miRNA Isolation Kit (Ambion), according to the manufacturer's instructions. cDNA was synthesized from total RNA samples using the iScript™ cDNA Synthesis Kit (Bio-Rad) or the Taq-Man™ MicroRNA Reverse Transcription Kit (Applied Biosystems) according to the manufacturer's instructions. qPCR was performed with the CFX96™ Real-Time PCR Detection System (BioRad) for each cDNA sample in technical duplicates. Relative mRNA levels were detected by SYBRGreen incorporation and calculated by the $2^{-\Delta\Delta Cq}$ method using *HPRT1* as internal reference. The absolute mRNA abundance of *ABCD1*, *ABCD2*, *BHRF1*, *EBNA1* and, for normalization, *HPRT1* were obtained by the SYBRGreen (*ABCD1*, *BHRF1*, and *EBNA1*) or TaqMan™ method (*ABCD2* and *HPRT1*) with quantification carried out using standard curves of known copy numbers obtained from linearized plasmids containing ABCD1, ABCD2, BHRF1, EBNA1 or HPRT cDNA. For reverse transcription and quantification of miR-155 and miR-9-5p, commercial TaqMan™ MicroRNA Assays were carried out according to the manufacturer's instructions (Applied Biosystems). Data was evaluated using Bio-Rad CFX-Manager and the $2^{-\Delta\Delta Ct}$ method after normalization with RNU6B as reference. Sequences of primers and TaqMan probes are listed in Supplementary Table 4.

**Luciferase reporter assay for evaluation of miRNA targets**. The full-length 3-UTR of the human ABCD1 (GenBank: NM_000033.3) or ABCD2 (GenBank: AJ000327.1) mRNA was cloned in the psiCHECK2 Renilla/Firefly dual-luciferase vector (Promega). For site-directed mutagenesis, the QuikChange Site Directed Mutagenesis Kit (Agilent) and primers listed in Supplementary Table 5 were used. HEK293 cells (CRL-1573™, ATCC) were co-transfected with the psiCHECK2 reporter plasmids and miRNA-mimics [miR-9-5p, miR-155 and negative control mimics lacking homology to human gene sequences (MISSION human miRNA mimics, Sigma-Aldrich)] in triplicates using Lipofectamine® 2000 (Invitrogen) according to the manufacturer's protocol. At 48 h post-transfection, Dual-Luciferase® Reporter Assays (Promega) were carried out using the GloMax®-Multi Detection System (Promega) and analyzed based on the ratio of Renilla and Firefly luciferase activities to normalize cell number and transfection efficiency.

**Western blotting**. Primary B cells before and 2 dpi or 5 dpi after EBV infection as well as EBV-immortalized B lymphocytes treated with either 25-HC (2 μM) or the solvent ethanol for 48 h were lysed in RIPA buffer containing protease inhibitors (Roche cOmplete), mixed with 5× sample buffer before being separated on a denaturing 7.5% polyacrylamide gel by discontinuous electrophoresis (SDS-PAGE), followed by semidry blotting onto nitrocellulose membrane. The blot was probed with primary mouse antibodies against the human ABCD1 protein (Euromedex ALD-1D6-AS, clone 2AL-1D6, 1:10,000) and β-actin (#603024087, Chemicon, 1:100,000) followed by a goat anti-mouse secondary antibody conjugated with horseradish peroxidase (#P0447, Dako, 1:30,000). For detection, the Immobilon Western HRP Substrate Peroxide Solution and Immobilon Reagent (Millipore) were used with the ChemiDoc Imaging System and Image Lab software (Bio-Rad).

**Bioinformatic analysis**. The full sequence of the 3′-UTR of the human ABCD1 (NM_000033.3) and ABCD2 (NM_005164.3) mRNAs were retrieved from the Entrez web portal. Potential binding sites for miRNA candidates were identified by comparing the results from different bioinformatics miRNA target prediction algorithms using the default settings of each tool including TargetScan Release 7.0 (https://www.targetscan.org)[58–61], miRanda-mirSVR[62] and PicTar (https://pictar.mdc-berlin.de/)[63]. Transcriptomics data were either directly analyzed using the GEO2R online tool with the default analysis parameter settings (significance level cut-off 0.05; Benjamin Hochberg correction for false discovery rate) or downloaded from the Gene expression omnibus (https://www.ncbi.nlm.nih.gov/geo/), and imported and analyzed using the Qlucore Software 3.5 (default analysis parameter settings: significance level cut-off 0.05; Two-Group comparison test).

**Statistics and reproducibility**. For statistical analysis, one-sided Fisher's exact test, two-tailed unpaired or paired, as well as nested Student's t-test, were used with Bonferroni adjustment to correct for multiple testing. *P*-values below 0.05 were regarded to indicate statistical significance. Graphs were produced and statistical results calculated using GraphPad 7.00.

**Reporting summary**. Further information on research design is available in the Nature Research Reporting Summary linked to this article.

## Data availability

The source data underlying Figs. 1–5 and Supplementary Figs. 2–7 are presented in the Supplementary Data file. Any other relevant data are available upon reasonable request. Unedited/uncropped western blot images are shown in Supplementary Fig. 8.

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

## Acknowledgements

We would like to thank all patients and healthy volunteers who participated in this study. This work was supported by the Austrian Science Fund KLI 837-B to I.W. and DOC 33-B27 to J.B. The authors have no conflicting financial interests.

## Author contributions

I.W. and J.B. designed the study. Data were collected by I.W., A.B., Z.P., M.T., S.H., and A.V.G. and analyzed by I.W., J.B., Z.P., and A.B. J.K. and J.H. performed the EBV-specific serology and contributed with discussion of the data. GC-Ms and ESI-MS data was contributed by G.R. and S.K., respectively. J.B. performed immunohistochemistry, S.H. performed in situ hybridization. F.E. contributed with the interpretation of data and F.E. and A.B.M. with EBV immortalization of B lymphocytes. M.K. and S.F.P. helped with interpretation of the data. I.W. created the figures and wrote; J.B., S.F.P., and M.K. edited the manuscript. All authors critically reviewed the manuscript and approved the final draft.

## Competing interests

The authors declare no competing interests.
