## [Peer Review File · Communications Biology]

Reviewers' comments:

Reviewer #1 (Remarks to the Author):

In this paper a previously reported relationship between herpesvirus infection and aberrant lipid metabolism, more specifically of very long chain fatty acids was further investigated. Novel findings are that the expression of the ABC transporters that translocate VLCFA across the peroxisomal membrane are reduced in EBV infected B-cells such that VLCFA cannot be degraded, the upregulated microRNAs were identified, and the authors showed that 25OH cholesterol can counteract these effects.

The paper is well written and the data are solid. Besides own experimental data, transcriptome and proteome databanks of EBV infected B-cells were reanalysed which reinforces the findings.

It was also investigated whether there is a link between EBV positivity and cerebral X-ALD but this did not seem to be the case. However in the abstract it is written "This is intriguing, as an unknown environmental factor triggers neuroinflammation in X-linked adrenoleukodystrophy (inherited ABCD1 dysfunction)". The way this is formulated is rather confusing as it leaves doubts on whether or not a link was found.

It is unclear why primary B cells of X-ALD patients do not accumulate VLCFA?

When printed on A4 paper, many titles of axes in the figures are too small and not readable.

How is the difference of the effect of 25HC in healthy and X-ALD lymphocytes interpreted?

Figure 5F: proliferation of peroxisomes is depicted in the scheme. This was not shown in the results. If it refers to a published paper, this should be mentioned.

Reviewer #2 (Remarks to the Author):

I read with great interest the manuscript entitled "Peroxisomal very long-chain fatty acid transport is targeted by herpesviruses and the antiviral host response" by Isabelle Weinhofer et al. In this manuscript, the authors report that EBV infection increases the levels of VLCFA by impairing their degradation via repression of the VLCFA transporters ABCD1 and ABCD2. EBV was shown to achieve this regulation by inducing the expression of the human miRNAs miR-9-5p and miR-155, resulting in increased VLCFA levels. Curiously, treatment with 25-HC was shown to restore ABCD1 expression and impair the production of virions. Altogether, the results of this study bring novel mechanisms that unravel unknown pathways related to both EBV viral infection and pathogenesis.

The authors present a very original hypothesis and employ adequate methods with transparent data representation and statistics. I commend the authors for the use of scatter plots and the overall graphic representation of the data.

I came across a few shortcomings and questions while reading this manuscript. In brief, the abstract and introduction have some sentences that are difficult to follow and need revision. The material and methods section lacks a few essential descriptions that will improve clarity and reproducibility. I also raised some questions regarding the methods, results, and overall findings of this study. I recommend this paper for publication upon completion/revision of the minor concerns outlined below.

Overall Concerns:

In this study, the levels of saturated VLCFAs from healthy control donors were compared against primary B-cells infected with EBV in vitro. The authors used TPA and butyrate to induce lytic induction in immortalized B-cells, filtering the supernatant and directly treating primary B-cells. Taking into account the described methodology:

1. Is it plausible to find TPA/butyrate contaminants in the medium containing viral particles added into primary B-cells? If positive,
2. Can phorbol esters such as TPA regulate pathways related to LCFA or VLCFA synthesis? If positive, can it be a source of bias?

Please provide a brief rationale for the questions raised above.

A. Abstract

"VLCFA, normally of low abundance ..." Low abundance where? Please specify.

"EBV infection interfered early with" consider replace to "EBV early infection interfered with..." or "EBV infection interfered with early expression of the VLCFA transporters...". This sentence is ambiguous.

"Application of 25-HC". Application is not the proper term here. Consider "Treatment with" or

"Intervention with."

"This is intriguing," consider rephrasing or changing to "These intriguing findings may be related to..."

B. Introduction

This section should be revised for clarity. There are a few sentences that are very difficult to read and follow, such as the sentence below:

"Endogenous synthesis of long- and very long-chain fatty acids is accomplished by ER-embedded enzymes including seven fatty acyl chain-selective elongases (ELOVL1-7) catalyzing the first, rate-limiting condensation step of the elongation cycle."

In addition to the hypothesis, please provide the objective or briefly describe how did you tackle the questions you have raised.

C. Results

Figure 1C : C20:0 and C24:4 have discrepancies and not enough data points. Why the experimental number is different across the assessment of LFCA and VLFA (n=3-7)? Were the experiments conducted in duplicates or represent a single time point collection?

Figure 2A and B: Graphs show "absolute" instead of relative mRNA levels. There is no indication that an absolute quantification was conducted to detect ABCD1 and ABCD2 levels. Please clarify.

EBV positivity is not elevated in post-mortem brain tissue of children with cerebral ALD: In Supplementary Figure 4 authors show that "no positive staining was detected for LMP1". Do you think that LMP1 is the best target to assess EBV positivity in brain tissues? Did the authors consider using EBER-ISH (gold standard) or even EBNA-1 antibody? The small sample size and inadequate detection method undermine the IHC findings and do not corroborate the overall findings of the CALD cohort. Please consider removing or providing a better rationale to keep IHC results.

D. Discussion

"It was tempting to speculate that the inherent ABCD1 deficiency in X-ALD patients would confer a higher risk for EBV infection and ... However, we did not detect a higher prevalence of EBV seropositivity in childhood CALD patients than in controls of similar age with conditions unrelated to X-ALD."

It seems like there is a particular age-specific prevalence of EBV infection among kids and young adults. For instance, this survey conducted in the US (PMID: 23868878) shows that the EBV antibody prevalence in children aged 6-8 years old was 50% while young adults aged 18-19 was 89%. Do you think there might be an analysis bias in the age-matched control groups of the CALD cohort?

E. Material and Methods

- Age of healthy volunteers from the X-ALD panel is provided as mean, while the median is used for the X-ALD arm. Please use uniform, descriptive statistics to describe participants in the study, including median and SD for age (e.g. n=; median age=, SD =).

What method was used to reverse transcribe miRNAs (i.e., in-house primers, commercial kits)? In the case of commercial kits, please provide adequate citations.

For the luciferase assays, what reagents or kits were used to determine renilla and firefly activity (Dual Glo Luciferase Assay Kit)?

In the bioinformatic analysis section, please provided (when possible) the accession links of the online platforms used (e.g., TargetScanHuman, Release 7.1, https://www.targetscan.org/vert_71/).

Throughout the M&M section:

1. The catalog number for certain antibodies is omitted. Please provided the cat number of all primary and secondary antibodies used for IHC, blots, and FACS, as well as the cat number of serological kits used.
2. Authors have not disclosed the model and specification of the equipment used for CLIA serological tests, scintillation counting, qPCR reactions, and luciferase assays.

Open Questions:

- Treatment with 25-HC was able to revert EBV-induced VLCFA and block EBV replication in vitro. Curiously, the levels of ABCD1 and ABCD2 are markedly reduced in early dpis and seem to increase over time.

1. Do you think that ABCD1 and ABCD2 expression is associated with the activation of the lytic cycle or the expression of EBV latent genes?
2. Since 25-HC can also interfere with EBV replication, have you planned to test other EBV lytic cycle inhibitors such as acyclovir or valpromide?
3. The gene expression of ABCD1 and ABCD2 was interestingly correlated with that of BHRF1 and EBNA1, showing an inverse correlation. What about assessing the gene expression of BZLF1 and gp350?

- The expression of the human miRNAs hsa-miR-155 and hsa-miR-9-5p were shown to be induced by EBV infection and target ABCD1 and ABCD2 expression. Did the authors also think of looking into viral miRNAs such as ebv-miR-BHRF1 and ebv-miR-BART?

Reviewer #3 (Remarks to the Author):

In this paper, Weinhofer et al., described EBV infection interfered early with expression of the VLCFA transporters ABCD1 and ABCD2, impairing VLCFA degradation within peroxisomes. microRNAs targets in ABCD1 and ABCD2 were also predicted for studying EBV-induced RNA interference. Targeting of ABCD1 was found to be shared by other herpesviruses and coronaviruses. Bioinformatic results were confirmed with lab works that enhanced these findings. However, I have some suggestions for the authors to improve this article as follows.

1. In the Supplementary Fig. 1, did the heatmap show Normalized counts of differently expressed genes between B cells and EBV-immortalized B-lymphocytes? If these are Normalized counts, what is the count normalization method used, such as RPKM and CMP? I found the values in the heatmap key ranges from -2 to 2. Did the heatmap was scaled with Z-Score or other methods? I would suggest authors to authors to includes information for these questions in the figure legend or method part. I

would also suggest authors to have a table for the P values and log fold changes of these genes showed on the heatmap.

2. On page 5, lines 146-148. Authors found no significant alterations with ELOVL1 and ELOVL7 and even downregulated expression of ELOVL3 using RT-qPCR. How are the expression levels of these genes in the transcriptome sequencing data?

3. It is same as the comment 1, I would suggest authors to provide the information of count normalization and scale for the heatmap in Fig. 2E. What is the count normalization method for the Fig. 2F?

4. On page 7, lines 202-204. Authors identified two putative miR-9-5p binding sites in the 3'-UTR of ABCD1 and three putative miR-155 binding sites in the 3'-UTR of ABCD2 used several miRNA target prediction tools. Are these binding sites were identified by all prediction tools used? I would also suggest authors to use the outputs of prediction tools as Supplementary information.

5. On page 7, lines 232-234. I would suggest authors have a Fisher's exact test for Supplementary Fig. 4a to look at the statistical significance.

6. In Fig. 4, authors used arbitrary unit as y-axis. How did this number was calculated from the transcriptomic read count? I would also suggest authors to show the fold change and p-value for each sample in fig. 4.

7. On page 15, "Bioinformatic analysis and statistics" in the method part. For repeatability of this project, could the authors indicate the parameters setting for the miRNA target prediction tools and GEO2R.

POINT-BY-POINT RESPONSE Weinhofer et al., “Peroxisomal very long-chain fatty acid transport is targeted by herpesviruses and the antiviral host response”

REVIEWERS’ COMMENTS

Reviewer #1 (Remarks to the Author):

In this paper a previously reported relationship between herpesvirus infection and aberrant lipid metabolism, more specifically of very long chain fatty acids was further investigated. Novel findings are that the expression of the ABC transporters that translocate VLCFA across the peroxisomal membrane are reduced in EBV infected B-cells such that VLCFA cannot be degraded, the upregulated microRNAs were identified, and the authors showed that 25OH cholesterol can counteract these effects.

The paper is well written and the data are solid. Besides own experimental data, transcriptome and proteome databanks of EBV infected B-cells were reanalysed which reinforces the findings.

It was also investigated whether there is a link between EBV positivity and cerebral X-ALD but this did not seem to be the case. However in the abstract it is written “This is intriguing, as an unknown environmental factor triggers neuroinflammation in X-linked adrenoleukodystrophy (inherited ABCD1 dysfunction)”. The way this is formulated is rather confusing as it leaves doubts on whether or not a link was found.

It is unclear why primary B cells of X-ALD patients do not accumulate VLCFA?

When printed on A4 paper, many titles of axes in the figures are too small and not readable.

How is the difference of the effect of 25HC in healthy and X-ALD lymphocytes interpreted?

Figure 5F: proliferation of peroxisomes is depicted in the scheme. This was not shown in the results.

If it refers to a published paper, this should be mentioned.

Reviewer #2 (Remarks to the Author):

I read with great interest the manuscript entitled “Peroxisomal very long-chain fatty acid transport is targeted by herpesviruses and the antiviral host response” by Isabelle Weinhofer et al. In this manuscript, the authors report that EBV infection increases the levels of VLCFA by impairing their degradation via repression of the VLCFA transporters ABCD1 and ABCD2. EBV was shown to achieve this regulation by inducing the expression of the human miRNAs miR-9-5p and miR-155, resulting in increased VLCFA levels. Curiously, treatment with 25-HC was shown to restore ABCD1 expression and impair the production of virions. Altogether, the results of this study bring novel mechanisms that unravel unknown pathways related to both EBV viral infection and pathogenesis.

The authors present a very original hypothesis and employ adequate methods with transparent data representation and statistics. I commend the authors for the use of scatter plots and the overall graphic representation of the data.

I came across a few shortcomings and questions while reading this manuscript. In brief, the abstract and introduction have some sentences that are difficult to follow and need revision. The material and methods section lacks a few essential descriptions that will improve clarity and reproducibility. I also raised some questions regarding the methods, results, and overall findings of this study. I recommend this paper for publication upon completion/revision of the minor concerns outlined

below.

Overall Concerns:

In this study, the levels of saturated VLCFAs from healthy control donors were compared against primary B-cells infected with EBV in vitro. The authors used TPA and butyrate to induce lytic induction in immortalized B-cells, filtering the supernatant and directly treating primary B-cells. Taking into account the described methodology:

1. Is it plausible to find TPA/butyrate contaminants in the medium containing viral particles added into primary B-cells? If positive,
2. Can phorbol esters such as TPA regulate pathways related to LCFA or VLCFA synthesis? If positive, can it be a source of bias?

Please provide a brief rationale for the questions raised above.

A. Abstract

"VLCFA, normally of low abundance ..." Low abundance where? Please specify.

"EBV infection interfered early with" consider replace to "EBV early infection interfered with..." or "EBV infection interfered with early expression of the VLCFA transporters...". This sentence is ambiguous.

"Application of 25-HC". Application is not the proper term here. Consider "Treatment with" or "Intervention with."

"This is intriguing," consider rephrasing or changing to "These intriguing findings may be related to..."

B. Introduction

This section should be revised for clarity. There are a few sentences that are very difficult to read and follow, such as the sentence below:

"Endogenous synthesis of long- and very long-chain fatty acids is accomplished by ER-embedded enzymes including seven fatty acyl chain-selective elongases (ELOVL1-7) catalyzing the first, rate-limiting condensation step of the elongation cycle."

In addition to the hypothesis, please provide the objective or briefly describe how did you tackle the questions you have raised.

C. Results

Figure 1C : C20:0 and C24:4 have discrepancies and not enough data points. Why the experimental number is different across the assessment of LFCA and VLFCFA (n=3-7)? Were the experiments conducted in duplicates or represent a single time point collection?

Figure 2A and B: Graphs show "absolute" instead of relative mRNA levels. There is no indication that an absolute quantification was conducted to detect ABCD1 and ABCD2 levels. Please clarify.

EBV positivity is not elevated in post-mortem brain tissue of children with cerebral ALD: In Supplementary Figure 4 authors show that "no positive staining was detected for LMP1". Do you think that LMP1 is the best target to assess EBV positivity in brain tissues? Did the authors consider using EBER-ISH (gold standard) or even EBNA-1 antibody? The small sample size and inadequate

detection method undermine the IHC findings and do not corroborate the overall findings of the CALD cohort. Please consider removing or providing a better rationale to keep IHC results.

D. Discussion

“It was tempting to speculate that the inherent ABCD1 deficiency in X-ALD patients would confer a higher risk for EBV infection and ... However, we did not detect a higher prevalence of EBV seropositivity in childhood CALD patients than in controls of similar age with conditions unrelated to X-ALD.”

It seems like there is a particular age-specific prevalence of EBV infection among kids and young adults. For instance, this survey conducted in the US (PMID: 23868878) shows that the EBV antibody prevalence in children aged 6-8 years old was 50% while young adults aged 18-19 was 89%. Do you think there might be an analysis bias in the age-matched control groups of the CALD cohort?

E. Material and Methods

- Age of healthy volunteers from the X-ALD panel is provided as mean, while the median is used for the X-ALD arm. Please use uniform, descriptive statistics to describe participants in the study, including median and SD for age (e.g. n=; median age=, SD =).

What method was used to reverse transcribe miRNAs (i.e., in-house primers, commercial kits)? In the case of commercial kits, please provide adequate citations.

For the luciferase assays, what reagents or kits were used to determine renilla and firefly activity (Dual Glo Luciferase Assay Kit)?

In the bioinformatic analysis section, please provided (when possible) the accession links of the online platforms used (e.g., TargetScanHuman, Release 7.1, https://www.targetscan.org/vert_71/).

Throughout the M&M section:

1. The catalog number for certain antibodies is omitted. Please provided the cat number of all primary and secondary antibodies used for IHC, blots, and FACS, as well as the cat number of serological kits used.
2. Authors have not disclosed the model and specification of the equipment used for CLIA serological tests, scintillation counting, qPCR reactions, and luciferase assays.

Open Questions:

- Treatment with 25-HC was able to revert EBV-induced VLCFA and block EBV replication in vitro. Curiously, the levels of ABCD1 and ABCD2 are markedly reduced in early dpis and seem to increase over time.

1. Do you think that ABCD1 and ABCD2 expression is associated with the activation of the lytic cycle or the expression of EBV latent genes?
2. Since 25-HC can also interfere with EBV replication, have you planned to test other EBV lytic cycle inhibitors such as acyclovir or valpromide?
3. The gene expression of ABCD1 and ABCD2 was interestingly correlated with that of BHRF1 and EBNA1, showing an inverse correlation. What about assessing the gene expression of BZLF1 and gp350?

- The expression of the human miRNAs hsa-miR-155 and hsa-miR-9-5p were shown to be induced by EBV infection and target ABCD1 and ABCD2 expression. Did the authors also think of looking into viral miRNAs such as ebv-miR-BHRF1 and ebv-miR-BART?

Reviewer #3 (Remarks to the Author):

In this paper, Weinhofer et al., described EBV infection interfered early with expression of the VLCFA transporters ABCD1 and ABCD2, impairing VLCFA degradation within peroxisomes. microRNAs targets in ABCD1 and ABCD2 were also predicted for studying EBV-induced RNA interference. Targeting of ABCD1 was found to be shared by other herpesviruses and coronaviruses. Bioinformatic results were confirmed with lab works that enhanced these findings. However, I have some suggestions for the authors to improve this article as follows.

1. In the Supplementary Fig. 1, did the heatmap show Normalized counts of differently expressed genes between B cells and EBV-immortalized B-lymphocytes? If these are Normalized counts, what is the count normalization method used, such as RPKM and CMP? I found the values in the heatmap key ranges from -2 to 2. Did the heatmap was scaled with Z-Score or other methods? I would suggest to authors to includes information for these questions in the figure legend or method part. I would also suggest authors to have a table for the P values and log fold changes of these genes showed on the heatmap.

2. On page 5, lines 146-148. Authors found no significant alterations with ELOVL1 and ELOVL7 and even downregulated expression of ELOVL3 using RT-qPCR. How are the expression levels of these genes in the transcriptome sequencing data?

3. It is same as the comment 1, I would suggest authors to provide the information of count normalization and scale for the heatmap in Fig. 2E. What is the count normalization method for the Fig. 2F?

4. On page 7, lines 202-204. Authors identified two putative miR-9-5p binding sites in the 3'-UTR of ABCD1 and three putative miR-155 binding sites in the 3'-UTR of ABCD2 used several miRNA target prediction tools. Are these binding sites were identified by all prediction tools used? I would also suggest authors to use the outputs of prediction tools as Supplementary information.

5. On page 7, lines 232-234. I would suggest authors have a Fisher's exact test for Supplementary Fig. 4a to look at the statistical significance.

6. In Fig. 4, authors used arbitrary unit as y-axis. How did this number was calculated from the transcriptomic read count? I would also suggest authors to show the fold change and p-value for each sample in fig. 4.

7. On page 15, "Bioinformatic analysis and statistics" in the method part. For repeatability of this project, could the authors indicate the parameters setting for the miRNA target prediction tools and GEO2R.

SPECIFIC COMMENTS TO REVIEWER #1

1) It was also investigated whether there is a link between EBV positivity and cerebral X-ALD but this did not seem to be the case. However in the abstract it is written “This is intriguing, as an unknown environmental factor triggers neuroinflammation in X-linked adrenoleukodystrophy (inherited ABCD1 dysfunction)”. The way this is formulated is rather confusing as it leaves doubts on whether or not a link was found.

We thank the reviewer for this valuable comment. In order to avoid confusion and to clarify the outcome of our findings, we have included the following changes:

Changes in the Manuscript: Abstract (line 50-55)

Because viral infection might trigger neuroinflammation in X-linked adrenoleukodystrophy (X-ALD, inherited ABCD1 deficiency), we explored a possible link between EBV-infection and cerebral X-ALD. However, neither immunohistochemistry of post-mortem brains nor analysis of EBV seropositivity in 35 X-ALD children supported involvement of EBV in the onset of neuroinflammation. Collectively, our findings indicate a previously unrecognized, pivotal role of ABCD1 in viral infection and host defence, prompting consideration of other viral triggers in cerebral X-ALD.

2) It is unclear why primary B cells of X-ALD patients do not accumulate VLCFA?

In previous work, we compared both the expression of *ABCD1* and accumulation of VLCFAs in primary, peripheral immune cells including monocytes, granulocytes, T cells and B cells isolated from X-ALD patients and healthy control (Weber et al., Human Molecular Genetics, 2014). In that study, we found that primary B cells from X-ALD patients do not accumulate VLCFAs and explained this by our observation that B cells express the ABCD1-related peroxisomal transporter ABCD2 to similar levels as ABCD1. ABCD1 and ABCD2 have overlapping substrate specificities and ABCD2 has been shown to functionally compensate for ABCD1-deficiency both *in vitro* and *in vivo* (Netik et al., Human Molecular Genetics 1999; Pujol et al., Human Molecular Genetics 2004).

Changes in the Manuscript: Introduction, p.4 (line 108-109)

Therefore, it was surprising that, in contrast to transformed B lymphocytes, primary B cells from X-ALD patients do not accumulate VLCFAs¹², which is probably due to relatively high expression of the related ABCD2 gene encoding an ABCD1 homolog with overlapping functions^{19,20}.

3) When printed on A4 paper, many titles of axes in the figures are too small and not readable.

We accordingly changed the size of figures in Fig. 2, Fig. 3, Fig. 4 and Fig. 5 as well as in Suppl. Fig. 6.

4) How is the difference of the effect of 25HC in healthy and X-ALD lymphocytes interpreted?

We thank the reviewer for perceiving the lack of discussion on this important aspect within our manuscript. Accordingly, we have incorporated the following change in the discussion section:

Changes in the Manuscript: Discussion, p.11 (line 351-357)

Because the VLCFA C26:0 has similar hydrophobicity and dissociation rate from membranes as cholesterol^{51,52}, it seems plausible that limiting also C26:0-containing lipids is necessary to make membranes less favourable for viral entry and/or release. Thus, one could hypothesize that the lack of efficacy of 25-HC on stimulating ABCD1 expression and thus, the inability to induce VLCFA transport and degradation in the context of ABCD1 deficiency, might interfere with the capacity of 25-HC to regulate the VLCFA-content in specific membrane domains required for the release of EBV virions from

infected cells. Future research needs to address in detail the underlying mechanism for how the lack of ABCD1 interferes with the ability of 25-HC to prevent EBV replication.

5) Figure 5F: proliferation of peroxisomes is depicted in the scheme. This was not shown in the results. If it refers to a published paper, this should be mentioned.

The depiction of peroxisome proliferation in the context of EBV-infection in the model displayed in Fig. 5F is not based on direct experimental evidence from this manuscript. To clarify that this hypothetical assumption is based on Fig. 2E showing the upregulation of genes encoding proteins involved in peroxisome biogenesis and proliferation such as the *peroxisomal biogenesis factors* *PEX3*, *PEX14*, *PEX11A/B* and *mitochondrial fission 1 (FIS1)*, normally correlating with increased peroxisome numbers, we included the following change in the manuscript:

Changes in the Manuscript: Legend Fig.5, p.25 (line 712-715)

(F) Model of EBV targeting VLCFA metabolism and peroxisomes (left panel) and of how 25-HC impedes viral reproduction (right panel). EBV infection of B cells results in rapid miRNA-mediated downregulation of the peroxisomal VLCFA importers ABCD1 and ABCD2. This occurs despite a general induction of peroxisome-related genes such as those encoding proteins involved in peroxisome proliferation and biogenesis (c.f. Fig. 2e), possibly indicating increased peroxisome numbers. The impaired peroxisomal degradation of VLCFAs, but concurrent upregulation of VLCFA synthesis via the fatty acid elongase ELOVL1, leads to increased cellular VLCFA levels, which may be necessary to establish latency.

SPECIFIC COMMENTS TO REVIEWER #2

In this study, the levels of saturated VLCFAs from healthy control donors were compared against primary B-cells infected with EBV in vitro. The authors used TPA and butyrate to induce lytic induction in immortalized B-cells, filtering the supernatant and directly treating primary B-cells.

Taking into account the described methodology:

- 1). Is it plausible to find TPA/butyrate contaminants in the medium containing viral particles added into primary B-cells? If positive,***
- 2). Can phorbol esters such as TPA regulate pathways related to LCFA or VLCFA synthesis? If positive, can it be a source of bias?***

Please provide a brief rationale for the questions raised above.

We would like to thank the reviewer for bringing up this important aspect of potential bias. Indeed, the phorbol ester TPA was previously found to positively regulate *de novo* fatty acid synthesis in human monocytic U937 cells (Xiong et al., JBC 2011). In this publication, Xiong et al. suggested as the underlying mechanism to be TPA-mediated elevation of biotin synthesis, which is required for acetyl-CoA carboxylase activity. Thus, TPA contaminations cannot be entirely excluded as source of bias for our results. However, increased ceramide concentrations, resulting from elevated synthesis of fatty acids, were only observed with 1 μ M TPA but not with 0.1 μ M. In our study, we used 0.03 μ M TPA to induce the lytic programme in immortalized B-lymphocytes. As only an aliquot of the supernatant corresponding to around 1×10^7 EBV particle was added to the primary B cells, any residual TPA would be even further diluted at this step. In addition, we observed that the levels of saturated long-chain FAs remained unchanged 5 dpi, which argues against a general induction of LCFA synthesis by the

presence of residual TPA in the medium of EBV-infected B cells. We searched the literature for putative effects of TPA on the elongation of VLCFAs but were unable to identify data supporting this concept.

Regarding sodium butyrate, previous work revealed a role in stimulating fatty acid oxidation and decreasing fatty acid synthesis (Moreau et al., Pharmacological Research 2019). Thus, these data would not support a putative effect of residual sodium butyrate in stimulating VLCFA levels upon infection of primary B cells with EBV.

A. Abstract

“VLCFA, normally of low abundance ...” Low abundance where? Please specify.

Due to the word count limitation set by *Communications Biology*, we had to shorten the abstract from 236 words to 200 words and thus, have removed the statement on the abundance of VLCFAs.

“EBV infection interfered early with” consider replace to “EBV early infection interfered with...” or “EBV infection interfered with early expression of the VLCFA transporters...”. This sentence is ambiguous.

Changes in the Manuscript: Abstract (line 40)

Gene expression profiling revealed that, despite a general induction of peroxisome-related genes, EBV early infection decreased expression of the peroxisomal VLCFA transporters ABCD1 and ABCD2, thus impairing VLCFA degradation.

“Application of 25-HC”. Application is not the proper term here. Consider “Treatment with” or “Intervention with.”

Changes in the Manuscript: Abstract (line 44)

Treatment with 25-hydroxycholesterol, an antiviral innate immune modulator produced by macrophages, restored ABCD1 expression and reduced VLCFA accumulation in EBV-infected B-lymphocytes, and, upon lytic reactivation, reduced virus production in control but not ABCD1-deficient cells.

“This is intriguing,” consider rephrasing or changing to “These intriguing findings may be related to...”

Changes in the Manuscript: Abstract (c.f. also Reviewer 1, point 1) (line 50-55)

Because viral infection might trigger neuroinflammation in X-linked adrenoleukodystrophy (X-ALD, inherited ABCD1 deficiency), we explored a possible link between EBV-infection and cerebral X-ALD. However, neither immunohistochemistry of post-mortem brains nor analysis of EBV seropositivity in 35 X-ALD children supported involvement of EBV in the onset of neuroinflammation. Collectively, our findings indicate a previously unrecognized, pivotal role of ABCD1 in viral infection and host defence, prompting consideration of other viral triggers in cerebral X-ALD.

B. Introduction

This section should be revised for clarity. There are a few sentences that are very difficult to read and follow, such as the sentence below:

To improve clarity we have accordingly modified the following sentences:

“Endogenous synthesis of long- and very long-chain fatty acids is accomplished by ER-embedded

enzymes including seven fatty acyl chain-selective elongases (ELOVL1-7) catalyzing the first, rate-limiting condensation step of the elongation cycle.”

Changes in the Manuscript: Introduction, p. 3 (line 84-87)

Endogenous synthesis of long- and very long-chain fatty acids is accomplished by ER-embedded fatty acyl chain-selective enzymes (elongation of very long chain fatty acids, ELOVL). The ELOVL family comprises of seven members (ELOVL1-7) which catalyze the first, rate-limiting condensation step of the elongation cycle.

In addition to the hypothesis, please provide the objective or briefly describe how did you tackle the questions you have raised.

We thank the reviewer for this suggestion and included the following changes in the manuscript:

Changes in the Manuscript: Introduction, p. 4 (line 113-118)

Here, we hypothesized that modulation of cellular VLCFA homeostasis could be a common strategy adopted by various (herpes)viruses to support their life cycle. Thus, using EBV as a paradigm for viral infections, the main objective of this study was to determine whether the peroxisomal import and degradation of VLCFAs is critical for viral infection and/or host defence. Our findings reveal a novel central role of ABCD1, a peroxisomal VLCFA transporter linked to the neuroinflammatory disorder X-ALD, in interfering with virus-mediated induction of VLCFA metabolism.

C. Results

Figure 1C : C20:0 and C24:4 have discrepancies and not enough data points. Why the experimental number is different across the assessment of LFCFA and VLFCFA (n=3-7)? Were the experiments conducted in duplicates or represent a single time point collection?

At the timepoint analysed, C20:0, C22:0, C24:0 and C26:0 are present at very low amounts in B cells when compared to C16:0 or C18:0 (mean concentration 0.27 nmol/mg protein; 2.19 nmol/mg protein; 1.29 nmol/mg protein and 0.12 nmol/mg protein vs. 66.01 nmol/mg protein and 46.37 nmol/mg protein, respectively). Further, only a limited number of B cells can be isolated from the blood of donors, and after infection, before proliferation is induced by the virus, the number of B cells is further diminished. Therefore, the detection of these fatty acids is technically very demanding and, thus, was carried out by a specialized lab at the Academic Medical Center, University of Amsterdam, The Netherlands, that is specialized at detecting VLCFAs in the context of X-ALD. Due to these limitations, we were unable to perform technical replicates for these samples. Thus, the experiment was carried out with B cells isolated from seven donors, with each data point representing the value of B cells derived from a single individual. We were able to measure C20:0 and C24:0 in three and four out of seven donors, respectively. For clarification, we have included the following change in our manuscript:

Changes in the Manuscript: Legend Fig.1, p.19 (line 609-611)

(C) The levels of saturated long-chain and very long-chain (\geq C22) fatty acids (FA) were determined by ESI-MS 5 days post infection of primary B cells from 7 healthy donors with EBV *in vitro* (MOI=10). For C20:0, C24:0 and C26:0, the quantification was limited to n=3, n=4 and n=6, respectively, due to low abundance and associated detection limits of these fatty acids. The data are depicted as boxplots (median \pm interquartile range). * $P \leq 0.05$, two-tailed nested Student's *t*-test.

Figure 2A and B: Graphs show “absolute” instead of relative mRNA levels. There is no indication that an absolute quantification was conducted to detect ABCD1 and ABCD2 levels. Please clarify.

We thank the reviewer for pointing out this lack of clarity within our manuscript. We have added the following change to the description of absolute and relative quantification in the methods section, paragraph “RNA isolation and reverse transcription-coupled PCR (RT-qPCR)” and accordingly changed the labelling of the y-axis in Fig. 2D.

Changes in the Manuscript: Materials and Methods, p. 15-16 (line 494-500)

qPCR was performed with the CFX96™ Real-Time PCR Detection System (BioRad) for each cDNA sample in technical duplicates. Relative mRNA levels were detected by SYBRGreen incorporation and calculated by the $2^{-\Delta\Delta Cq}$ method using *HPRT1* as internal reference. The absolute mRNA abundance of *ABCD1*, *ABCD2*, *BHRF1*, *EBNA1* and, for normalization, *HPRT1* were obtained by the SYBRGreen (*ABCD1*, *BHRF1* and *EBNA1*) or TaqMan™ method (*ABCD2* and *HPRT1*) with quantification carried out using standard curves of known copy numbers obtained from linearized plasmids containing *ABCD1*, *ABCD2*, *BHRF1*, *EBNA1* or *HPRT* cDNA. For quantification of miR-155 and miR-9-5p, TaqMan™ MicroRNA Assays were carried out following the recommended protocol (Applied Biosystems).

Changes in the Manuscript: Fig. 2D, p.20

EBV positivity is not elevated in post-mortem brain tissue of children with cerebral ALD: In Supplementary Figure 4 authors show that “no positive staining was detected for LMP1”. Do you think that LMP1 is the best target to assess EBV positivity in brain tissues? Did the authors consider using EBER-ISH (gold standard) or even EBNA-1 antibody? The small sample size and inadequate detection method undermine the IHC findings and do not corroborate the overall findings of the CALD cohort. Please consider removing or providing a better rationale to keep IHC results.

We thank the reviewer for this valuable suggestion and agree that with regard to our limited sample size, ISH using the EBV-associated RNA EBER, a highly abundant non-protein coding small RNA expressed in all forms of latency, is more meaningful than IHC against LMP1, which is expressed only during the growth and default programmes of the virus. Accordingly, we additionally performed EBER-ISH on post-mortem brain tissue of the 4 CALD cases. However, also with this methodology we were unable to detect EBV positivity in the brain tissue of these patients despite positivity found in post-mortem brain tissue of a case with lymphomatoid granulomatosis serving as positive control.

Changes in the Manuscript: Supplementary Fig. 4F

Supplementary Figure 4. EBV positivity is not elevated in serum and post-mortem brain tissue of children with cerebral ALD. (A-D) Sera from children with proven CALD before allogeneic stem cell transplantation ($n=35$, age range= 4-15 years, median age=8 years) and from controls with conditions unrelated to X-ALD ($n=285$, age range= 4-15 years, median age=8 years) were tested for EBV seropositivity using antibodies directed against the viral capsid antigen (VCA) IgG and IgM; n.s.= non significant (one-sided Fisher's exact test) (A). The distribution of age within the group of controls and CALD patients is shown in (B). The data are depicted as boxplots (median \pm interquartile range). In (C), both control and CALD groups are splitted up according to age and the EBV positivity is shown for each of these age subgroups. (D) The positive samples were assessed for past, acute or recent EBV infections using antibodies directed against VCA IgG, VCA IgM and EBV nuclear antigen (EBNA)-1 IgG (B). (E) Immunohistochemistry for the B cell marker CD20 and the EBV protein LMP1 (latent membrane protein 1) and (F, G) in situ hybridization for the EBV-associated small RNA EBV was performed on post-mortem brain tissue of 4 CALD cases. One representative case with B cell infiltration of

perivascular cuffs is shown (E, F). No positive staining was detected for either LMP1 or EBER. For the EBER in situ hybridization, post-mortem brain tissue of a case with lymphomatoid granulomatosis served as positive control (G).

Changes in the Manuscript: Materials and Methods, p. 13 (line 427-433)

EBER in situ hybridization

EBER in situ hybridization was performed using a commercially available kit. Briefly, formalin-fixed, paraffin-embedded sections were hybridized with the Inform EBER probe (#05278660001, Roche Diagnostics) and the VENTANA ISH iView Blue Detection Kit was used for visualization (#05278511001, Roche Diagnostics). The staining procedures were performed in a Ventana BenchMark ULTRA IHC/ISH System (Roche Diagnostics) according to guidelines from the supplier.

Changes in the Manuscript: Results, p.8 (line 241-243)

Finally, by using immunohistochemistry for detection of EBV latent membrane protein 1 (LMP1) and in situ hybridization for the EBV-associated small non-coding RNA EBER, we were unable to identify EBV positivity in post-mortem brain tissue of four CALD cases (Supplementary Fig. 4e,f).

D.Discussion

“It was tempting to speculate that the inherent ABCD1 deficiency in X-ALD patients would confer a higher risk for EBV infection and ... However, we did not detect a higher prevalence of EBV seropositivity in childhood CALD patients than in controls of similar age with conditions unrelated to X-ALD.”

It seems like there is a particular age-specific prevalence of EBV infection among kids and young adults. For instance, this survey conducted in the US (PMID: 23868878) shows that the EBV antibody prevalence in children aged 6-8 years old was 50% while young adults aged 18-19 was 89%. Do you think there might be an analysis bias in the age-matched control groups of the CALD cohort?

We are grateful to the reviewer for bringing up this important aspect. Indeed, we had extensive internal discussions about the rate of seropositivity in the control group and the question of a potential analysis bias. The age-matched control group consisted of sera from 285 patients (age range 4 – 15 years, median age=8) derived from the oncology department at the Charité Universitätsmedizin Berlin (Germany). These control patients received medical treatment for reasons unrelated to CALD. Furthermore, no patients were included that were tested because of suspicion for EBV or other viral infections. Both control and CALD samples were measured at the Charité Universitätsmedizin Berlin using the same protocol. When compared to the overall rate of EBV seropositivity of 50% for children (n=472, age 5-8 years) in the United States, as determined by Balfour et al, the observed incidence of 73% (n=142, age 6-8 years) or 70% (n=285, age range 4-15 years, median age=8 years) in our control group indeed seems to be quite high. As factors such as race/ethnicity are known to influence the rate of seropositivity, we did a literature search to identify the current EBV prevalence in children in Berlin/Germany. In the study of Abrahamyan et al, J Neurol Neurosurg Psychiatry 2020, the authors compared the EBV seropositivity in MS patients of a large hospital population (n=16,163) in Berlin. The cohort consisted of sera from persons that were included retrospectively irrespective of diagnosis or the reason for ordering EBV serology. Within the age group of 5-9 years (n=577), the authors revealed a seropositivity of 68% and within the group of 10-14 years (n=712), a seropositivity of 71% was identified. Accordingly, the revealed EBV prevalence of 70% determined for our control group seems to be comparable to the EBV prevalence determined by Abrahamyan and colleagues for children of this age group in Berlin/Germany.

To enable more precise comparison with the literature and to prevent misconception on this important aspect, we introduced the following changes:

Changes in the Manuscript: Results, p. 8 (line 233-236)

EBV serum positivity is not elevated in children with inflammatory cerebral X-ALD

To test the hypothesis that EBV constitutes a trigger for the onset of inflammatory cerebral ALD, we investigated the presence of EBV seropositivity in a group of 35 children with MRI-confirmed CALD lesions (age range 4 – 15 years, median age = 8 years). Our analysis revealed similar incidences in the X-ALD (63%) and the age-matched control groups (70%) consisting of 285 children (age range 4 – 15 years, median age = 8 years) with conditions unrelated to X-ALD (Supplementary Fig. 4a).

Changes in the Manuscript: Supplementary Fig. 4 A-C

Supplementary Figure 4. EBV positivity is not elevated in serum and post-mortem brain tissue of children with cerebral ALD. (A-D) Sera from children with proven CALD before allogeneic stem cell transplantation ($n=35$, age range= 4-15 years, median age=8 years) and from controls with conditions unrelated to X-ALD ($n=285$, age range= 4-15 years, median age=8 years) were tested for EBV seropositivity using antibodies directed against the viral capsid antigen (VCA) IgG and IgM; n.s.= non significant (one-sided Fisher's exact test) (A). The distribution of age within the group of controls and CALD patients is shown in (B). The data are depicted as boxplots (median \pm interquartile range). In (C), both control and CALD groups are splitted up according to age and the EBV positivity is shown for each of these age subgroups.

Changes in the Manuscript: Discussion, p. 12 (line 378-381)

However, we did not detect a higher prevalence of EBV seropositivity in childhood CALD patients than in controls of similar age with conditions unrelated to X-ALD. Of note, the determined prevalence of EBV infection in our cohort of control children aged 4-15 years was comparable to rates previously observed for children within this age group in Berlin/Germany⁵⁴, which was higher than the incidence reported in a U.S. study⁵⁵.

E. Material and Methods

- Age of healthy volunteers from the X-ALD panel is provided as mean, while the median is used for the X-ALD arm. Please use uniform, descriptive statistics to describe participants in the study, including median and SD for age (e.g. $n=$; median age=, SD =).

Changes in the Manuscript: Materials and Methods, p. 13 (line 399-400)

The study included peripheral blood samples from 5 adult X-ALD patients (median age=38 years, SD=4.12) and 20 healthy volunteers (median age=37 years, SD=10.64) for isolation of B cells.

What method was used to reverse transcribe miRNAs (i.e., in-house primers, commercial kits)? In the case of commercial kits, please provide adequate citations.

To clarify, we included the following change in the manuscript:

Changes in the Manuscript: Materials and Methods, p. 16 (line 498-500)

For reverse transcription and quantification of miR-155 and miR-9-5p, commercial TaqMan™ MicroRNA Assays were carried out according to the manufacturer's instructions (Applied Biosystems).

For the luciferase assays, what reagents or kits were used to determine renilla and firefly activity (Dual Glo Luciferase Assay Kit)?

Changes in the Manuscript: Materials and Methods, p. 16 (line 512-513)

At 48 h post-transfection, Dual-Luciferase® Reporter Assays (Promega) were carried out and analyzed based on the ratio of Renilla and Firefly luciferase activities to normalize cell number and transfection efficiency.

In the bioinformatic analysis section, please provided (when possible) the accession links of the online platforms used (e.g., TargetScanHuman, Release 7.1, https://www.targetscan.org/vert_71/).

Changes in the Manuscript: Materials and Methods, p. 17 (line 537-538)

Potential binding sites for miRNA candidates were identified by comparing the results from different bioinformatics miRNA target prediction algorithms including TargetScan Release 7.0⁵⁵⁻⁵⁸ (<https://www.targetscan.org>), miRanda-mirSVR⁵⁹ and PicTar⁶⁰ (<https://pictar.mdc-berlin.de/>).

Throughout the M&M section:

1. The catalog number for certain antibodies is omitted. Please provided the cat number of all primary and secondary antibodies used for IHC, blots, and FACS, as well as the cat number of serological kits used.

Changes in the Manuscript: Materials and Methods, p. 13 (line 413-416)

The serologic testing was performed by detection of IgG and IgM directed to the viral capsid protein (VCA, antigen used: p18; REF 310510 and REF 310500, respectively, DiaSorin S.p.A., Saluggia, Italy) and IgG to the EB nuclear antigen-1, applying LIAISON® EBNA IgG (REF 310520, DiaSorin S.p.A., Saluggia, Italy). Chemiluminescent immunoassays were carried out using a LIAISON XL (DiaSorin S.p.A., Saluggia, Italy).

Changes in the Manuscript: Materials and Methods, p. 13 (line 423-424)

Immunohistochemical analysis was performed on formalin-fixed, paraffin-embedded sections using primary antibodies for CD20 (Thermo scientific, MS-340, 1:100) and EBV-LMP (Dakocytomation, M0897, 1:60). Visualization of bound primary antibody was performed by incubation with Biotin-labelled donkey anti-mouse secondary antibodies (#715-065-150, Jackson ImmunoResearch Laboratories, West Grove, PA) for 1 h at room temperature and detection with avidin-labelled peroxidase (Sigma, 1:150) and diaminobenzidine tetrachloride (DAB, Sigma). A hematoxylin counterstain was performed to visualize nuclei.

Changes in the Manuscript: Materials and Methods, p. 14 (line 439-441)

The purity of isolated primary B cells was determined using a mouse anti-human CD19 monoclonal antibody, PE conjugated (clone LT19, #130-091-247, Miltenyi) and the PE-conjugated monoclonal antibody isotype control (#120-002-723, Miltenyi) by flow cytometry as described previously¹².

Changes in the Manuscript: Materials and Methods, p. 16 (line 524-526)

The blot was probed with primary mouse antibodies against the human ABCD1 protein (Euromedex ALD-1D6-AS, clone 2AL-1D6, 1:10,000) and β -actin (#603024087, Chemicon, 1:100,000) followed by a goat anti-mouse secondary antibody conjugated with horseradish peroxidase (#P0447, Dako, 1:30,000).

2. Authors have not disclosed the model and specification of the equipment used for CLIA serological tests, scintillation counting, qPCR reactions, and luciferase assays.

Changes in the Manuscript: Materials and Methods, p. 13 (line 413-414)

The serologic testing was performed by detection of IgG and IgM directed to the viral capsid protein (VCA, antigen used: p18; REF 310510 and REF 310500, respectively, DiaSorin S.p.A., Saluggia, Italy) and IgG to the EB nuclear antigen-1, applying LIAISON[®] EBNA IgG (REF 310520, DiaSorin S.p.A., Saluggia, Italy). Chemiluminescent immunoassays were carried out using a LIAISON XL (DiaSorin S.p.A., Saluggia, Italy).

Changes in the Manuscript: Materials and Methods, p. 15 (line 482-483)

After protein precipitation by HClO₄, a Folch partition was carried out and ¹⁴C-acetate was determined in the aqueous phase by scintillation counting using the Perkin Elmer Tri-Carb 4910TR Scintillation Counter.

Changes in the Manuscript: Materials and Methods, p. 16 (line 512-513)

At 48 h post-transfection, Luciferase assays were carried out using the GloMax[®]-Multi Detection System (Promega) and analyzed based on the ratio of Renilla and Firefly luciferase activities to normalize cell number and transfection efficiency.

Open Questions:

- Treatment with 25-HC was able to revert EBV-induced VLCFA and block EBV replication in vitro. Curiously, the levels of ABCD1 and ABCD2 are markedly reduced in early dpis and seem to increase over time.

The data retrieved from the transcriptomics study of Mrozek-Gorska and colleagues (Mrozek-Gorska et al., PNAS 2019) using B cells derived from 3 donors revealed only a transient decrease in *ABCD1* and *ABCD2* expression with normalized levels by 4 dpi. In contrast, our own data derived from B cells (n=7) and B-lymphocytes (n=11) indicated persistent downregulation of *ABCD1* and *ABCD2*. In addition, also data retrieved from the microarray study involving B cells and B lymphocytes from 6 donors by Caliskan et al (HumMolGen 2011) revealed sustained repression of *ABCD1* upon EBV-immortalization. However, both our study and the microarray data from Caliskan et al. indicate some yet unexplained biological/donation derived variability between different individuals in the level of *ABCD1* and *ABCD2* expression.

1. Do you think that ABCD1 and ABCD2 expression is associated with the activation of the lytic cycle or the expression of EBV latent genes?

Ersing and colleagues previously reported a temporal proteomic map of EBV B cell lytic replication (Ersing et al., CellReports 2017). Using the interactive worksheet “Plotter” in Table S1 of that publication, we analyzed how induction of the EBV lytic replication in type II EBV+ Burkitt lymphoma and type I EBV+ Akata cells impact ABCD1 and ABCD2 expression. Whereas ABCD2 was not present in this data set, we found that ABCD1 protein levels remained unchanged by the onset of lytic EBV replication. Thus, this data does not support a link between ABCD1 expression and activation of the lytic cycle. We added the following paragraph to the Discussion section:

Changes in the Manuscript: Discussion, p. 11 (line 357-362)

With our data linking ABCD1 function to antiviral properties of 25-HC in the context of EBV replication, the question raised whether ABCD1 expression might be associated with activation of the EBV lytic cycle in infected B cells. However, using an interactive worksheet previously published by Ersing and colleagues, who reported a temporal proteomic map of EBV B cell lytic replication⁵³, we found no support linking a change in ABCD1 protein levels to induction of EBV lytic replication.

2. Since 25-HC can also interfere with EBV replication, have you planned to test other EBV lytic cycle inhibitors such as acyclovir or valpromide?

We would like to thank the reviewer for the suggestion to test in addition to 25-HC whether another EBV lytic cycle inhibitor may interfere with ABCD1 expression. Accordingly, we treated healthy control B lymphocytes (n=4) for 24 hrs with valpromide (10 mM) and investigated ABCD1, ABCD2 and also ELOVL1 expression by qPCR. We found that similar to 25-HC, also valpromide upregulated ABCD1 expression in the treated cells. In contrast to 25-HC, however, valpromide also significantly stimulated ABCD2 expression but did not change ELOVL1 mRNA levels. We added these findings as Supplementary Figure 7 to the manuscript.

Changes in the Manuscript: New Supplementary Figure 7

Supplementary Figure 7. The EBV lytic cycle inhibitor valpromide stimulates ABCD1 and ABCD2 expression. EBV-immortalized B lymphocytes derived from healthy donors (n=4) were treated with 10 mM valpromide or the solvent DMSO for 24 h before RNA was isolated and RT-qPCR carried out for ABCD1, ABCD2 and ELOVL1 expression. Data were normalized to HPRT mRNA levels and are displayed as fold induction over DMSO-treated cells. The mean is indicated by a horizontal line ± S.D. For statistical analysis, the raw values used to generate the fold-change display were used (two-tailed paired Student’s t-test, *P<0.05).

Changes in the Manuscript: Results, p. 9 (line 292-298)

Collectively, these data show that the antiviral mediator 25-HC restores *ABCD1* expression in B lymphocytes, resulting in reduced VLCFA levels and an *ABCD1*-dependent decrease in EBV production (Fig. 5f). Finally, we tested whether the impact on *ABCD1* and *ELOVL1* expression is shared by other compounds known to interfere with EBV replication. Accordingly, we analyzed the expression of *ABCD1*, *ELOVL1* and also *ABCD2* in EBV-immortalized healthy control B lymphocytes treated with the EBV lytic cycle inhibitor valpromide for 24 h. Our data revealed that similar to 25-HC, valpromide significantly upregulated *ABCD1* mRNA levels and additionally stimulated *ABCD2* expression in EBV-immortalized B lymphocytes. However, in contrast to 25-HC, valpromide had no significant effect on *ELOVL1* transcription (Supplementary Fig. 7).

Changes in the Manuscript: Discussion, p. 11 (line 341-343)

Our finding that the antiviral cholesterol metabolite 25-HC promotes cellular VLCFA degradation by positive modulation of *ABCD1* expression with simultaneous downregulation of *ELOVL1* to curb VLCFA synthesis lends support to this concept. Of note, the ability of 25-HC to stimulate *ABCD1* expression was shared by the EBV lytic cycle inhibitor valpromide, possibly indicating a more general mechanism in targeting *ABCD1* to interfere with EBV replication.

3. The gene expression of *ABCD1* and *ABCD2* was interestingly correlated with that of *BHRF1* and *EBNA1*, showing an inverse correlation. What about assessing the gene expression of *BZLF1* and *gp350*?

Our intention to measure *BHRF1* and *EBNA1* expression was to provide a positive control for successful infection of primary B cells with EBV. To even further verify this aspect, we additionally investigated the expression of immediate-early viral gene *BZLF1* and included this data in Fig. 2D.

Changes in the Manuscript: Figure 2D, p.20

Figure 2. Expression dynamics of genes related to peroxisomes and VLCFA synthesis during EBV infection. (D) Expression profile of primary B cells before and after EBV infection. RNA was isolated from primary B cells of healthy donors ($n=3-5$) before and at one or two days post infection (dpi) with EBV *in vitro*. RT-qPCR was carried out for the *ABCD1* and *ABCD2* genes with viral *EBNA1*, *BHRF1* and *BZLF1* serving as controls for successful infection. *HPRT* was used for normalization purposes. The bar graphs show mean \pm S.D. of the indicated values. For statistical analysis, two-tailed ratio paired Student's *t*-test was used (* $P \leq 0.05$, ** $P \leq 0.01$).

Changes in the manuscript: Supplementary Table 1

Supplementary Table 1. Primers used for RT-qPCR and EBV genome copy number analysis

Gene	Accession number, mRNA	Product length (bp)	Sequence
HPRT	NM_000194	220	F 5'-ccctggcgtcgtgattagt-3' R 5'-caggtcagcaaagaatttatagcc-3' FAM-caggactgaacgtcttgctcgaga-BHQ1
ABCD1	NM_000033	169	F 5'-gagaacatccccatcgtc-3' R 5'-tgtagagcacaccaccgta-3'
ABCD2	NM_005164	79	F 5'-tcctacacaatgtccatctct-3' R 5'-aggacatctttccagtcca-3' Cy5-caaagagaaggaggatgggatgc-BHQ2
ABCD3	M81182	98	F 5'-cggctcatcacaacagtga-3' R 5'-aggtgttcaccagttttcg-3'
EBNA1	NC_007605	217	F 5'-gccggtgtgttcgtatatgg-3' R 5'-ccttcaaacctcagcaaatataga-3'
BHRF1	NC_007605	208	F 5'-ggagatactgttagccctg-3' R 5'-gtgtgtataaatctgttccaag-3'
GNPAT	NM_014236	176	F 5'-tcagaaactacaaagagccatcca-3' R 5'-ttcgtagcagctcaccaacc-3'
FAR1	NM_032228	243	F 5'-gacagacaccacaagagcga-3' R 5'-tcacatttaactgaacagcatcttta-3'
AGPS	NM_003659	131	F 5'-aggggaaggaatgtttgagcga-3' R 5'-acactgttcctccaccaattg-3'
ELOVL1	NM_022821	279	F 5'-attgagctgatggacacagtgat-3' R 5'-gaccaggacaaaactggatcagc-3'
ELOVL3	NM_152310	64	F 5'- agcaaggctatagaactcggagac-3' R 5'- taaagatgagtgccgcttacg-3'
ELOVL7	NM_024930	77	F 5'- ggccagcctaccagaagtatttg-3' R 5'- ggcgacaataacaaactggacaag-3'
BALF5	MK_540419 (EBV genome)	193	F 5'-agcttgatgacgatgccaca-3' R 5'-aggatggaaagggcatgtgg-3'
BZLF1	MK973062 (EBV Genome)	113	F 5'-ctgttggtttccgtgtgc-3' R 5'-cagtgggtttgcttggccc-3'

-The expression of the human miRNAs hsa-miR-155 and hsa-miR-9-5p were shown to be induced by EBV infection and target ABCD1 and ABCD2 expression. Did the authors also think of looking into viral miRNAs such as ebv-miR-BHRF1 and ebv-miR-BART?

We highly appreciate this suggestion of Reviewer 2 to investigate whether next to cellular miRNAs also virus-encoded miRNAs such as ebv-miR-BHRF1 and ebv-miR-BART would target *ABCD1* or *ABCD2*. Accordingly, we screened the mirTar database (Shu et al, cells 2018; <https://mcube.nju.edu.cn/jwang/mirTar/docs/mirTar/>), which is a comprehensive miRNA target repository that includes bidirectional interspecies actions between humans and viruses. Regarding *ABCD1*, we found 61 candidate target sites for viral miRNAs that included next to herpesvirus HSV-1, HSV-2 and HCMV-derived miRNAs also the ebv-miR-BART10-5p from EBV. Of note, hcmv-miR-US29-3p, one of the HCMV miRNAs proposed by the mirTar database to target *ABCD1*, was verified by Kim et al, Cell Host Microbe 2015, in their landscape study to identify Host-Virus crosstalk during HCMV infection. For *ABCD2*, the programme predicted 12 targets including a putative binding site for the ebv-miR-BART1-5p. We included these findings in the discussion section.

Changes in the manuscript: Discussion, p. 10 (line 318-324)

Lending support to this concept, a screen for host-virus interactions of the mirTar database (<https://mcube.nju.edu.cn/jwang/mirTar/docs/mirTar/>) revealed 61 putative interactions of virus miRNAs with *ABCD1* and 12 with *ABCD2*, including predicted binding sites for the EBV miRNAs ebv-miR-BART10-5p and ebv-miR-BART1-5p in the *ABCD1* and *ABCD2* mRNAs, respectively. Further studies are needed to confirm the functionality of these predicted host-virus interactions.

SPECIFIC COMMENTS TO REVIEWER #3

1. In the Supplementary Fig. 1, did the heatmap show Normalized counts of differently expressed genes between B cells and EBV-immortalized B-lymphocytes? If these are Normalized counts, what is the count normalization method used, such as RPKM and CMP? I found the values in the heatmap key ranges from -2 to 2. Did the heatmap was scaled with Z-Score or other methods? I would suggest authors to authors to includes information for these questions in the figure legend or method part. I would also suggest authors to have a table for the P values and log fold changes of these genes showed on the heatmap.

We thank the reviewer for the suggestion to incorporate the details concerning normalization and heatmap scaling as well as to calculate *P*-values and log fold changes. Using the Qlucore Omics Software we have accordingly redrawn the heatmap using the median levels for all probes of a specific gene, scaled the heatmap using Z-scores and calculated *P*-values and log₂-fold changes.

Changes in the manuscript: Supplementary Figure 1.

C

Peroxisomal genes:

Gene symbol	P-value	Log ₂ Fold Change
ABCD1	5.80E-16	-0.91
ABCD3	8.81E-12	0.43
ACAA1	1.14E-04	-0.27
ACAD11	9.08E-10	0.69
ACOT4	2.25E-18	1.79
ACOX2	2.54E-06	-0.26
ACOX3	3.49E-11	0.65
ACSF3	6.80E-03	-0.16
ACSL1	1.07E-02	-0.49
ACSL4	3.11E-02	0.10
ACSL5	1.32E-12	0.41

AGPS	1.57E-20	0.87
ALDH3A2	5.18E-07	0.45
AMACR	3.14E-13	0.39
CAT	4.85E-17	-0.91
CRAT	3.66E-02	-0.14
CROT	1.01E-13	0.53
DNM1L	2.68E-07	0.49
ECH1	7.11E-17	0.69
ECI2	4.25E-23	1.30
EHHADH	1.41E-13	0.43
FAR2	3.10E-03	0.40
FIS1	1.37E-11	-0.72
GNPAT	3.44E-13	0.52
GSTK1	8.67E-03	-0.35
HACL1	3.09E-10	0.80
HMGCL	1.38E-27	1.06
HSD17B4	1.31E-04	0.32
IDH1	1.59E-28	2.33
IDI1	1.15E-04	-0.69
ISOC1	1.51E-22	1.05
LDHA	1.05E-07	0.83
LONP2	2.32E-02	0.11
MDH1	1.39E-30	1.52
MLYCD	3.66E-17	0.56
MPV17	1.68E-12	0.53
PECR	6.72E-09	0.78
PEX11B	1.90E-42	1.92
PEX13	1.97E-04	0.15
PEX14	1.61E-10	0.48
PEX16	2.15E-15	-0.52
PEX2	1.79E-11	0.38
PEX3	2.82E-02	0.09
PEX6	4.54E-03	-0.36
PHYH	2.05E-15	0.81
PIPOX	1.86E-03	0.14
PMVK	6.09E-06	0.44
PRDX1	5.17E-36	2.06
PRDX5	1.80E-10	0.55
PXMP2	3.84E-12	1.08
PXMP4	3.47E-07	0.23
RHOC	6.24E-13	0.69
SCP2	8.34E-07	0.21
SLC25A17	6.96E-05	0.31
SLC27A2	1.44E-04	0.74
SOD1	1.68E-18	0.61
TMEM135	1.02E-14	0.72
TRIM37	4.29E-10	0.42

Genes encoding enzymes involved in LCFA and VLCFA synthesis:

Gene symbol	P-value	Log2 Fold Change
ELOVL1	1.76E-04	-0.25
ELOVL2	5.29E-01	-0.03
ELOVL3	3.78E-02	-0.08
ELOVL4	1.83E-01	0.10
ELOVL5	5.83E-05	-0.58
ELOVL6	4.40E-35	1.68
ELOVL7	7.91E-01	0.01

Supplementary Figure 1. Effect of EBV transformation on peroxisome related genes and expression of enzymes involved in LCFA and VLCFA synthesis. (A-B) mRNA levels from primary B cells derived from six healthy donors and their corresponding EBV-immortalized B lymphocytes were retrieved from the Gene Expression Omnibus database file GSE26212 and analyzed using the Qlucore Omics Explorer Software (Qlucore AB, Lund, Sweden). The heatmaps showing (A) significantly dysregulated peroxisomal genes (Two-Group comparison test, $P \leq 0.05$) and (B) genes encoding enzymes involved in LCFA and VLCFA synthesis were generated using the median value of all probes for the specific genes and scaled with Z-score. The transcriptomics data were generated by Caliskan *et al.*¹ by hybridization of the cDNAs to HumanHT-13 v3 Expression BeadChip arrays (Illumina Inc.). B cells are shown in two technical replicates; for the corresponding EBV-immortalized B lymphocytes, six independent cell lines were generated from each individual and analyzed in two technical replicates. The intensity estimates of the transcriptomics data were log-transformed and the quantile normalized using the “lumi” package in R v2.10.1 by Caliskan et al. ABCD1 is indicated by an arrow. P-values and log2 fold changes of the transcriptomics data are shown in (C).

2. On page 5, lines 146-148. Authors found no significant alterations with ELOVL1 and ELOVL7 and even downregulated expression of ELOVL3 using RT-qPCR. How are the expression levels of these genes in the transcriptome sequencing data?

We agree with the reviewer that including the expression data of the ELOVL enzymes involved in VLCFA synthesis retrieved from the transcriptomics study by Caliskan et al. would improve clarity and enable comparison of the data. We accordingly incorporated these data showing significant downregulation of ELOVL1 and unchanged ELOVL3 and ELOVL7 mRNA levels as Supplementary Fig. 1b,c (cf. changes in our response to comment 1, Reviewer 3).

3. It is same as the comment 1, I would suggest authors to provide the information of count normalization and scale for the heatmap in Fig. 2E. What is the count normalization method for the Fig. 2F?

To further improve the clarity of Fig. 2E, we have redrawn the heatmap using log-transformed data and also included hierarchical clustering. In addition, we added to the figure legend that data normalized by Mrozek-Gorska for sequencing depth using size factors was imported into Qlucore Omics Explorer, where data was log-transformed and scaled with Z-Score.

Changes in the manuscript: Figure 2E, p.20

Figure 2. Expression dynamics of genes related to peroxisomes and VLCFA synthesis during EBV infection. (A, B)(E-F) Time-resolved RNA-Seq data from B cells, isolated from three healthy donors and infected *in vitro* with EBV, was retrieved from Mrozek-Gorska et al. ²². Data normalized by Mrozek-Gorska and colleagues for sequencing depth using size factors were imported into the Qlucore Omics Explorer, where log-transformation and scaling with Z-score was carried out. Heatmap and graphs of peroxisome-related genes (E) and elongase encoding genes involved in saturated LCFA and VLCFA synthesis (F) at different time points following infection. Depicted data points are means of three donors \pm S.D.

4. On page 7, lines 202-204. Authors identified two putative miR-9-5p binding sites in the 3'-UTR of ABCD1 and three putative miR-155 binding sites in the 3'-UTR of ABCD2 used several miRNA target prediction tools. Are these binding sites were identified by all prediction tools used? I would also suggest authors to use the outputs of prediction tools as Supplementary information.

We agree with the reviewer that implementing the outputs of the prediction tools within the manuscript would enable the reader to compare and assess the predicted target sites within the human *ABCD1* and *ABCD2* 3'-UTRs. For *ABCD1*, five different bioinformatic miRNA target prediction algorithms (TargetScan v7.0, PicTar, DIANA-microT-CDS v5.0, miRCode 11 and miRDB) predicted the same two miR-9-5p target sites, located at nucleotide positions 533-538 and 862-868 after the stop codon, within the 3'-UTR of *ABCD1*. For *ABCD2*, one of three miR-155 target sites were predicted by five prediction tools (TargetScan v7.0, microCosm targets v5, miRCode 11, miRanda-mirSVR and miRDB). Only the miRanda-mirSVR ¹³ algorithm predicted two additional miR-155 target sites in the 3'-UTR of *ABCD2*. We have now added this information in two new Supplementary tables (Supplementary Tables 3 and 4).

Changes in the manuscript: New Supplementary Tables 3 and 4

Supplementary Table 3. Prediction analysis for miR-9-5p targeting *ABCD1*

Database	Number of predicted miR-9-5p target sites	Nt-Position after the stop codon	Score
TargetScan v7.0 https://www.targetscan.org/vert_70/	2	533-539; 862-868	Context++ score: -0.03 Context++ score: -0.12
PicTar https://pictar.mdc-berlin.de/	2	533, 862	PicTar score: 4.43
RNA22 v2 https://cm.jefferson.edu/rna22/Interactive/	2	421-439; 847-868	p -value: 0.043 p -value: 0.026
DIANA-microT-CDS v5.0 https://dianalab.e-ce.uth.gr/	3	388-405 523-538 843-867	Score: 0.0025 Score: 0.0052 Score: 0.0330
mirCode11 http://www.mircode.org/	2	533-538; 862-867	-
miRDB http://mirdb.org/	2	533, 862	Target score: 69

Five bioinformatic miRNA target prediction algorithms (TargetScan v7.0³⁻⁶, PicTar⁷, DIANA-microT-CDS v5.0^{8,9}, mirCode 11¹⁰ and miRDB^{11,12}) predicted the same two miR-9-5p target sites located at nucleotide positions 533-538 and 862-868 after the stop codon within the 3'-UTR of *ABCD1*. The second site, at position 862-868, was also predicted by the RNA22 v2¹³ algorithm, which also identified an additional putative site at position 421-439. Next to these two sites at nucleotide positions 533-538 and 862-868, the DIANA-microT-CDS vs5.0 algorithm predicted an additional miR-9-5p target site at position 388-405.

Supplementary Table 4. Prediction analysis for miR-155 targeting *ABCD2*

Database	Number of predicted miR-155 target sites	Nt-Position after the stop codon	Score
TargetScan v7.0 https://www.targetscan.org/vert_70/	1	128-134	Context++score: -0.02
microCosm targets v5 https://www.ebi.ac.uk	1	128-134	Rank metric score: 4,86
miRCode 11 http://www.mircode.org/	1	128-134	-
miRanda-mirSVR	3	128-134; 177-182; 195-199	mirSVR score: -1.126 mirSVR score: -0.039 mirSVR score: -0.456
miRDB http://mirdb.org/	1	128	Target score: 75
RNA22 v2 https://cm.jefferson.edu/rna22/Interactive/	-		
PicTar https://pictar.mdc-berlin.de/	-		

Five bioinformatic miRNA target prediction algorithms (TargetScan v7.0³⁻⁶, microCosm targets v5¹⁴, miRCode 11¹⁰, miRanda-mirSVR¹³ and miRDB^{11,12}) predicted the same miR-155 target site located at nucleotide position 128-135 after the stop codon within the 3'-UTR of *ABCD2*. The miRanda-mirSVR¹³ algorithm predicted two additional miR-155 target sites in the human *ABCD2* 3'-UTR. The RNA22 v2¹³ and PicTar⁷ algorithms did not identify any putative miR-155 target site within the 3'-UTR of *ABCD2*.

5. On page 7, lines 232-234. I would suggest authors have a Fisher's exact test for Supplementary Fig. 4a to look at the statistical significance.

We thank the reviewer for this suggestion to do a Fisher's exact test for Supplementary Fig. 4A. As the alternative hypothesis states that the fraction of EBV-positive CALD children is higher than those of the control group, we used a one-sided Fisher's exact test. With a calculated odds ratio of 0.72 and a *P*-value of 0.8596, the alternative hypothesis was rejected. We have added the statistics to the manuscript:

Changes in the manuscript: Supplementary Fig. 4A

Supplementary Figure 4. EBV positivity is not elevated in serum and post-mortem brain tissue of children with cerebral ALD. (A-D) Sera from children with proven CALD before allogeneic stem cell transplantation (*n*=35, age range= 4-15 years, median age=8 years) and from controls with conditions unrelated to X-ALD (*n*=285, age range= 4-15 years, median age=8 years) were tested for EBV seropositivity using antibodies directed against the viral capsid antigen (VCA) IgG and IgM; n.s.= non significant (one-sided Fisher's exact test) (A).

6. In Fig. 4, authors used arbitrary unit as y-axis. How did this number was calculated from the transcriptomic read count? I would also suggest authors to show the fold change and p-value for each sample in fig. 4.

The transcriptomic data set was imported into Qlucore Software 3.5, used for analysis and to generate graphs. The arbitrary units on the y-axis in Qlucore are normalized variable values that were obtained using the default settings to Z-score normalization (mean zero and standard deviation 1). We agree with the reviewer that including fold changes and *P*-values for Fig. 4 would improve the clarity of the presented data. We accordingly added this information using a new Supplementary Table 5.

Changes in the manuscript: Legend Figure 4, p.24

Figure 4. ABCD1 expression is targeted by herpes- and coronaviruses. Transcriptomics datasets from herpes- or coronavirus *in vitro*-infected human cells (-, mock-infected; +, virus infected) or from peripheral blood mononuclear cells (PBMCs) derived from COVID-19 patients were retrieved from Gene Expression Omnibus database files. The data are depicted as boxplots (median ± interquartile range). **(A)** HSV-1 infected primary fibroblasts 9 h post infection (pi) (MOI=10, two biological replicates, GSE129582); **(B)** VZV infected melanoma cells, 36 h pi (two biological replicates, GSE85493); **(C)** EBV infected gastric cancer cells, 48 h pi (5 replicates, GSE135644); **(D)** HCMV infected lung fibroblasts, 72 h pi (MOI=10, two biological replicates, GSE99454); **(E)** roseolovirus infected T lymphoblastoid cells, 72 h pi (MOI=20, one biological replicate, GSE149808); **(F)** KSHV infected Tert-immortalized microvascular endothelial cells, 48 h pi (three replicates, GSE27136); **(G)** MERS-CoV infected bronchial epithelial cells, 24 h pi (MOI=5, three replicates, GSE45042); **(H)** SARS-CoV-1 infected bronchial epithelial cells, 3 h pi

(MOI=5, three replicates, GSE33267); **(I)** SARS-CoV-2 infected iPSC-cardiomyocytes, 72 h pi (MOI=0.1, three replicates, GSE150392); **(J)** PBMCs derived from three COVID-19 patients and three controls (CRA002390, <https://bigd.big.ac.cn/>). Arbitrary units on the y-axis represent normalized variable values that were obtained using the default settings to Z-score normalization in the Qlucore Software 3.5 (mean zero and standard deviation 1). For data sets including longitudinal sampling, time courses are shown in Supplementary Fig. 5. Log₂ fold changes and P-values are shown in Supplementary Table 5.

Changes in the manuscript: Legend Supplementary Figure 5

Supplementary Figure 5. Targeting of *ABCD1* expression by herpes- and coronaviruses at different time points. Time resolved transcriptomics datasets from human cells infected *in vitro* with herpes- and coronaviruses were retrieved from Gene Expression Omnibus database files and analyzed for *ABCD1* expression before and at different time points after infection. **(A)** VZV-infected melanoma cells (two biological replicates, GSE85493); **(B)** HCMV-infected lung fibroblasts (two biological replicates, GSE99454); **(C)** MERS-CoV-infected bronchial epithelial cells (three replicates, GSE45042); **(D)** SARS-CoV-1-infected bronchial epithelial cells (three replicates, GSE33267). The data are depicted as boxplots (median ± interquartile range). Arbitrary units on the y-axis represent normalized variable values that were obtained using the defaults to Z-score normalization in the Glucore Software 3.5 (mean 0 and standard deviation 1).

Changes in the manuscript: New Supplementary Table 5

Supplementary Table 5. P-values and Log₂ fold changes for *ABCD1* expression from transcriptomic datasets shown in Fig. 4

Data set	virus	P-value	Log ₂ Fold Change	Number of replicates per condition
GSE129582	HSV-1	8,88E-03	-1.80	2
GSE85493	VZV	3,02E-02	-0.37	2
GSE135644	EBV	1,75E-05	-0.48	5
GSE99454	HCMV	3,87E-04	-0.85	2
GSE149808	Roseolovirus	N.A.	N.A.	1
GSE27136	KSHV	3,77E-01	-0.07	3
GSE45042	MERS-CoV	8,13E-05	-2.13	3
GSE33267	SARS-CoV-1	3,55E-02	-0.24	3
GSE150392	SARS-CoV-2	1,03E-01	-0.69	3
CRA002390	SARS-CoV-2	6.96E-06*	1.38*	3

*P-value and log₂ fold change were obtained from Xiong et al. ¹⁵, Supplementary File 1 .

7. On page 15, “Bioinformatic analysis and statistics” in the method part. For repeatability of this project, could the authors indicate the parameters setting for the miRNA target prediction tools and GEO2R.

We accordingly added the following changes to the manuscript:

Changes in the manuscript: Materials and Methods, p. 17 (line 536-543)

Bioinformatic analysis, statistics and reproducibility

The full sequence of the 3'-UTR of the human ABCD1 (NM_000033.3) and ABCD2 (NM_005164.3) mRNAs were retrieved from the Entrez web portal. Potential binding sites for miRNA candidates were identified by comparing the results from different bioinformatics miRNA target prediction algorithms using the default settings of each tool including TargetScan Release 7.0 (<https://www.targetscan.org>)⁵⁶⁻⁵⁹, miRanda-mirSVR⁶⁰ and PicTar (<https://pictar.mdc-berlin.de/>)⁶¹. Transcriptomics data were either directly analyzed using the GEO2R online tool with the default analysis parameter settings (significance level cut-off 0.05; Benjamin Hochberg correction for false discovery rate) or downloaded from the Gene expression omnibus (<https://www.ncbi.nlm.nih.gov/geo/>), and imported and analyzed using the Qlucore Software 3.5 (default analysis parameter settings: significance level cut-off 0.05; Two-Group comparison test). For other statistical analysis, one-sided Fisher's exact test, two-tailed unpaired or paired as well as nested Student's *t*-test were used with Bonferroni adjustment to correct for multiple testing. *P*-values below 0.05 were regarded to indicate statistical significance. Graphs were produced and statistical results calculated using GraphPad 7.00.

REVIEWERS' COMMENTS:

Reviewer #1 (Remarks to the Author):

The authors addressed my remarks adequately. I don not have further questions.

Reviewer #2 (Remarks to the Author):

Thanks for resubmitting a new version of the manuscript entitled "Peroxisomal very long-chain fatty acid transport is targeted by herpesviruses and the antiviral host response" by Isabelle Weinhofer et al."

Overall, The authors addressed all questions regarding potential biases in results and discussion sections with sounding rationale and additional investigation.

There is a substantial improvement in the Material and Methods section. The authors now provide catalog numbers for reagents/equipment and an adequate description of methods.

Additional results were also conducted and presented in the new version of this manuscript. For instance, in addition to IHC for LMP1, the authors provide EBER in-situ hybridization of 4 CALD cases that substantiate their previous EBV-positivity findings. BLZF1 was added to the panel of positive controls for EBV infection in primary B-cells. The target genes ABCD1, ABCD2, and ELOVL1 were also evaluated using Valpromide, disclosing potential regulatory roles for ABCD1 and ABCD2 associated with EBV lytic cycle.

Results in the discussion section are also better articulated and present a more critical appraisal of the literature, including direct citations to other relevant papers that substantiate and strengthen the current findings.

In conclusion, this reviewer feels that his questions were adequately addressed and recommends this paper for publication.

Brunno F. R. Caetano, PhD.
Sao Paulo State University, Brazil

Reviewer #3 (Remarks to the Author):

Authors have address most low my questions. I am looking forward to the publication.